# D4: Improving LLM Pretraining via Document De-Duplication and Diversification

**Kushal Tirumala***
Meta AI Research

**Daniel Simig***
Meta AI Research

**Armen Aghajanyan**
Meta AI Research

**Ari S. Morcos**
Meta AI Research

## Abstract

Over recent years, an increasing amount of compute and data has been poured into training large language models (LLMs), usually by doing one-pass learning on as many tokens as possible randomly selected from large-scale web corpora. While training on ever-larger portions of the internet leads to consistent performance improvements, the size of these improvements diminishes with scale, and there has been little work exploring the effect of data selection on pre-training and downstream performance beyond simple de-duplication methods such as MinHash. Here, we show that careful data selection (on top of de-duplicated data) via pre-trained model embeddings can speed up training (20% efficiency gains) and improves average downstream accuracy on 16 NLP tasks (up to 2%) at the 6.7B model scale. Furthermore, we show that repeating data intelligently consistently *outperforms* baseline training (while repeating random data performs worse than baseline training). Our results indicate that clever data selection can significantly improve LLM pre-training, calls into question the common practice of training for a single epoch on as much data as possible, and demonstrates a path to keep improving our models past the limits of randomly sampling web data.

## 1 Introduction

Due to computational limits, initial work on language model pre-training focused on training models on small, high-quality text datasets such as BookCorpus [61] and Wikipedia [32]. More recently, however, catalyzed by works like [40], advancements in large language models (LLMs) have been driven by leveraging large collections of unlabeled, uncurated data derived from snapshots of the internet (CommonCrawl [41, 16, 39]), trading off small quantities of heavily-curated data for huge quantities of less-curated data. Because of the dramatic increase in data quantity, these strategies have resulted in higher performance models and have sparked a new paradigm wherein massive, largely unfiltered datasets are utilized for training [11, 50, 46].

Despite the essential role that large-scale web data now play in LM pre-training, data curation and selection for large-scale web data have not been thoroughly explored. This is primarily due to the universality of compute and data scaling laws [25, 20] which give practitioners a low-risk way to reliably improve LM performance by merely adding "more" data, not necessarily the "right" data. Indeed, the data selection method used to model scaling laws (along with the data selection methods used in most LLM pre-training pipelines) involves simply randomly sampling tokens from web data dumps that have been put through a combination of simple heuristic filtering (e.g., to eliminate very short strings) and very near match de-duplication [27].

If we continue relying on scaling laws to improve LLMs, we will quickly hit diminishing returns due to the power-law nature of scaling laws. We will therefore need exponentially more data to maintain a consistent marginal improvement, which may prove especially challenging as we are fast

---

*Equal contribution. Correspondence emails: ktirumala@meta.com, simigd@gmail.com

37th Conference on Neural Information Processing Systems (NeurIPS 2023) Track on Datasets and Benchmarks.

approaching the limits of available human-generated text data [51]. Encouragingly, in the context of vision, Sorscher et al. [47] demonstrated that we could leverage simple data selection strategies to overcome costly power-law scaling. They compare numerous data selection methods and find that clustering data points in a pre-trained embedding space and ranking according to the distance to the cluster centroid ("SSL Prototypes") significantly improves the data efficiency of vision models. Recently, Abbas et al. [1] demonstrated that using a pre-trained embedding space to de-duplicate data ("SemDeDup") improves both efficiency and performance of vision-language models such as CLIP. However, there has been little exploration of these or related approaches in training LLMs at scale. Motivated by this, we argue that by combining these approaches and applying them to LLMs, relatively simple data selection strategies leveraging pre-trained embeddings can significantly improve LLM training. Specifically, our contributions are as follows:

- We investigate different data selection strategies for standard LLM pre-training setups where data has already been manually filtered / de-duplicated (e.g., MinHash), and where we do not know the target distribution for which we optimize performance. We argue that the performance of SSL Prototypes is affected by duplicate-driven clusters in the embedding space. In Section 3.4 we propose a new data selection strategy **D4** that utilizes SemDeDup to avoid getting impacted by such clusters.

- In Section 4.1, we show that in the *compute-limited regime* where we have "infinite" source data and train models with fixed token budgets, we can achieve better pre-training perplexity and downstream accuracy than random iid data selection and previously established methods. Furthermore, we show that our method D4 can achieve around 20% efficiency gains at the 6.7b model scale, and that the magnitude of efficiency gains increases with model scale.

- In the *data-limited regime*, where we run out of data and must epoch over data, cleverly choosing what data to repeat can beat training on randomly selected new data, whereas randomly choosing data to repeat underperforms adding new data (Section 4.2). This calls into question the standard practice of single epoch LLM training, and suggests that epoching over intelligently subselected data might be a better approach.

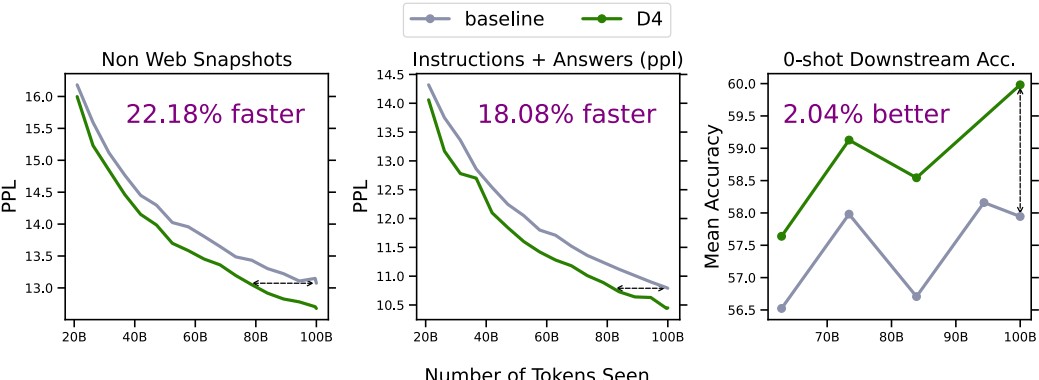

Figure 1: Learning curves for 6.7B OPT model pretraining on 100B tokens, with data selected with D4 (pink line) and randomly (gray line). D4 significantly outperforms baseline training, getting between 18-20% efficiency gains on validation perplexity and 2% increase in average 0-shot downstream accuracy across 16 NLP tasks. See Section A.2 for full learning curves.

## 2   Related Work

**Data selection in non-text domains:** Numerous works have successfully used data selection techniques in vision models [6, 10, 23, 31, 34, 38, 49], though these have largely been at sub-ImageNet scale. Some of these works develop pruning metrics that score individual data points (for example, EL2N from Paul et al. [38]), while some focus on data-efficiency and attempt to find groups of points that allow models to reach baseline performance with less data points, e.g., coresets [9, 35, 44, 60]. Sorscher et al. [47] compares many of the existing individual-score methods at ImageNet scale, finding that their SSL prototypes metrics and the (prohibitively expensive)

memorization metric from Feldman and Zhang [15] generally outperforms other methods. In the audio domain, Dong et al. [14] computes importance embeddings to find important training samples for audio scene classification. More recently, Abbas et al. [1] demonstrated very encouraging results on vision-language models (CLIP models) using SemDeDup — a similar method to SSL prototypes but focused on semantic deduplication. Our work combines these approaches and applies them to large-scale LLMs.

**Effect of pre-training data on LM performance:** Gao et al. [16] trains variants of GPT-2 [40] models from scratch to compare the "Pile" dataset to CommonCrawl-derived corpora. Radford et al. [40] demonstrates the positive impact of the quality filters and data de-duplication methods used to curate MassiveWeb by training 1.4B parameter models from scratch. Hernandez et al. [19] quantifies the effect of various amounts of artificially created data duplication and provides analysis on interpreting the changes in the behaviour of the models trained on duplicated data. Concurrently to our work, Xie et al. [56] propose using importance resampling to align the distribution of web data to high-quality reference corpora such as Wikipedia. Similarly, Gururangan et al. [17] explores data selection strategies for adapting LMs to a task-specific corpus. Another line of recent work explores how data mixture affects pre-training, with Xie et al. [55] demonstrating impressive improvements in downstream accuracy and perplexity across all datasets for 8B parameter models trained on the Pile. Similarly, Longpre et al. [30] explores the role of text quality, toxicity, age, and domain distribution of training data on LLM performance. Outside of data curation, there has been a recent surge of work exploring the impact of repeating data [5, 37, 57], generally concluding that repeating tokens is worse than training on new tokens (which we question in Section 4.2).

# 3 Experimental Setup

**Notation** : Given a source dataset, $D_{source}$, of documents (crawled web pages) and model architecture, $M$, we aim to find a strategy $S$ for selecting a subset of these documents that maximizes some evaluation metric $E(M(D_{S,R}))$. $R$ indicates the proportion of remaining documents from the source dataset $D_{source}$ after selecting data with strategy $S$. For this reason, we refer to $R$ throughout this work as the *selection ratio*: for example, if $R = 0.25$ and $|D_{source}| = 100$ million, then we *select* 25% of documents from a source dataset of size 100M documents to arrive at a a training dataset with 25M documents. We operate at the granularity of a single document, independently of how the model trainer would pack these documents into batches later. Throughout the paper, we use random selection as the baseline for $S$, as it is the most common method for selecting data for language model pre-training. In the rest of this section, we describe our choices of source dataset ($D_{source}$), model ($M$), evaluation metric ($E$), and, most importantly, our suggestions for the selection strategy ($S$).

## 3.1 Training Dataset (choice for $D_{source}$)

We perform all of our training runs on a version of CommonCrawl pre-processed with a CCNet [54] pipeline identical to the one used by Touvron et al. [50]. We add an additional step of MinHash-based de-duplication (see more details in Section A.1). Applying this common step before our experiments guarantees that any effects observed in our experiments complement the currently prevalent approach of MinHash-based data de-duplication strategies. Throughout the rest of this work, we refer to this dataset as *CC-dedup*.

## 3.2 Model Training (choices for $M$ and $T_{target}$)

To evaluate different configurations of data selection strategies, we train OPT [59] models from scratch on the pruned versions of datasets. We use the standard model architectures and settings of Zhang et al. [59] and use MetaSeq [59] to train all our models. For 125M models, we train to $T_{target} = 3B$ tokens. For 1.3B parameter models, we train to target token count of $T_{target} = 40B$. For 6.7B parameter models, we train to $T_{target} = 100B$ tokens. We choose these by trimming down the token budgets suggested by Hoffmann et al. [20] to meet our compute limitations. We provide full details of our training setup in Section A.1.

## 3.3 Evaluation Metrics (choices for $E$)

We keep most of our evaluation consistent with the setup from Zhang et al. [59].

**Validation Set Perplexity**. Our validation sets mainly come from [59], which includes validation sets derived from subsets of the Pile [16] such as CommonCrawl, DM Mathematics, HackerNews, OpenSubtitles, OpenWebText2, Project Gutenberg, USPTO, Wikipedia. We also include a validation set obtained from the PushShift.io Reddit dataset [4] (which we refer to as *redditflattened*). In addition, we measure perplexity on a validation set obtained from a train-validation split of our source dataset *CC-dedup*, and a validation set from C4 [41].

We notice that the effects of data selection vary significantly on individual validation sets depending on whether the validation set was derived from a web data corpus or not (see more details and analysis in Section 4.4.1). Motivated by this, we split validation sets into Web-snapshots (C4, CommonCrawl, and CC-dedup) and Non-web snapshots, and report average perplexity within these sets.

**Downstream Task Accuracy.** To evaluate downstream performance of our trained models, we report average 0-shot accuracy across the 16 NLP tasks from Zhang et al. [59], and use a prompting methodology consistent with Zhang et al. [59]. These set of 16 NLP tasks include Arc Challenge and ArcEasy [12], HellaSwag [58], OpenBookQA [33], PIQA [7], StoryCloze [36], Winograd [28], Winogrande [42], as well as tasks from SuperGLUE [52]. We refer the reader to Zhang et al. [59] for more information about this evaluation setup.

**Instruction Tuning Perplexity**. The evaluation mentioned above metrics presents an inherent trade-off. Though accuracy on downstream tasks is typically viewed as a more concrete representation of a language model's real-world value, its variance tends to be higher due to the limited number of examples in these tasks and the step-wise behavior of accuracy as a metric. In contrast, perplexity, as a metric, is smoother while still exhibiting a strong correlation with performance [43]. Therefore as a middle ground between the two evaluation metrics, we propose evaluating the perplexity on a sample drawn from the instruction-tuning dataset used for fine-tuning OPT-IML [21]. This dataset spans over 1500 unique NLP tasks and comprises a wide array of prompt-answer pairs and therefore is representative of the *average* NLP task. It has been carefully crafted by merging extensive task collections such as Super-NaturalInstructions [53] and PromptSource [3]. We refer the reader to Table 2.1 in [21] for a comprehensive breakdown. This approach allows us to balance practical performance measures and statistical consistency in evaluation. We note that this metric can simply be considered as perplexity on another validation set, where the validation set is filled with examples used for instruction-tuning (we are **not** fine-tuning on this dataset).

## 3.4 Data Selection Strategies (choices for $S$)

In our initial exploration of un-curated web data, we embedded a large sample of web documents, clustered these embeddings, and manually inspected the resulting clusters. We quickly identified several high density clusters with documents that had little to do with the natural distribution of human language and were artifacts of the web crawling: for example, advertisements of Nike shoes that were automatically generated from a single underlying template with minor modifications (see Section A.9 for details).

Motivated by the intuition that these duplicate-driven clusters need tshould be pruned, as well as the recent success of pruning methods in vision and vision-language models [1, 47], we focus our efforts on data selection strategies that manipulate data points based on their position in an embedding space. We embed each document by feeding it into a 125M OPT model and use the last-layer embedding of the last token (we experiment with different embedding spaces in Section A.7). Following this, we experiment with several approaches:

**SemDeDup**: Abbas et al. [1] proposed de-duplicating in both text and image domains by first using K-Means to cluster the embedding space, and removing points in each cluster that are within epsilon-balls of one another. We use this algorithm without any modifications and refer the reader to Abbas et al. [1] for implementation details of this algorithm.

**Prototypicality**: Sorscher et al. [47] investigated a large variety of data pruning strategies to improve the data efficiency of training image classification models, including a newly introduced "SSL Prototypes" metric that proved to be one of their best methods. This strategy involves first clustering the embedding space using k-means clustering and discarding data points in increasing order of their distance to the nearest cluster centroid, such that the most "prototypical" data points are discarded, enriching the much higher variance outliers. We refer the reader to Sorscher et al. [47] for a more detailed description of this algorithm.

**D4**: As mentioned previously, we find many instances of duplicate-driven clusters: clusters of templated text or extremely semantically redundant information that are not removed by MinHash. These regions of embedding space tend to be very dense and cause k-means to waste valuable cluster assignments on duplicated text. This biased clustering could also negatively to impact the effectiveness of SSL Prototypes since many clusters will be entirely driven by duplicates instead of more topical coherence. This insight lead us to our proposed strategy:

1. Apply *SemDeDup* with a selection ratio $R_{dedup}$ on the entire dataset $D$, producing a smaller dataset $D'$

2. Cluster points in $D'$ with K-Means

3. Apply *SSL Prototypes* on $D'$, with a selection ratio $R_{proto}$

The above-described strategy has an overall selection ratio of $R = R_{dedup} * R_{proto}$ and intends to diversify the distribution of our data locally and globally. For brevity we refer to this method as **D4**, a shorthand for *Document De-Duplication and Diversification*. Throughout this work, we choose $R_{dedup} = 0.75$ and vary $R_{proto}$ (we discuss this choice in Section A.1). In Section 4, we compare the performance of D4 to baseline training and other methods, and in Section 4.4 we analyze D4 and show that reclustering after semantic de-duplication indeed reduces the impact of duplicate-driven clusters (see Figure 7).

## 4 Results

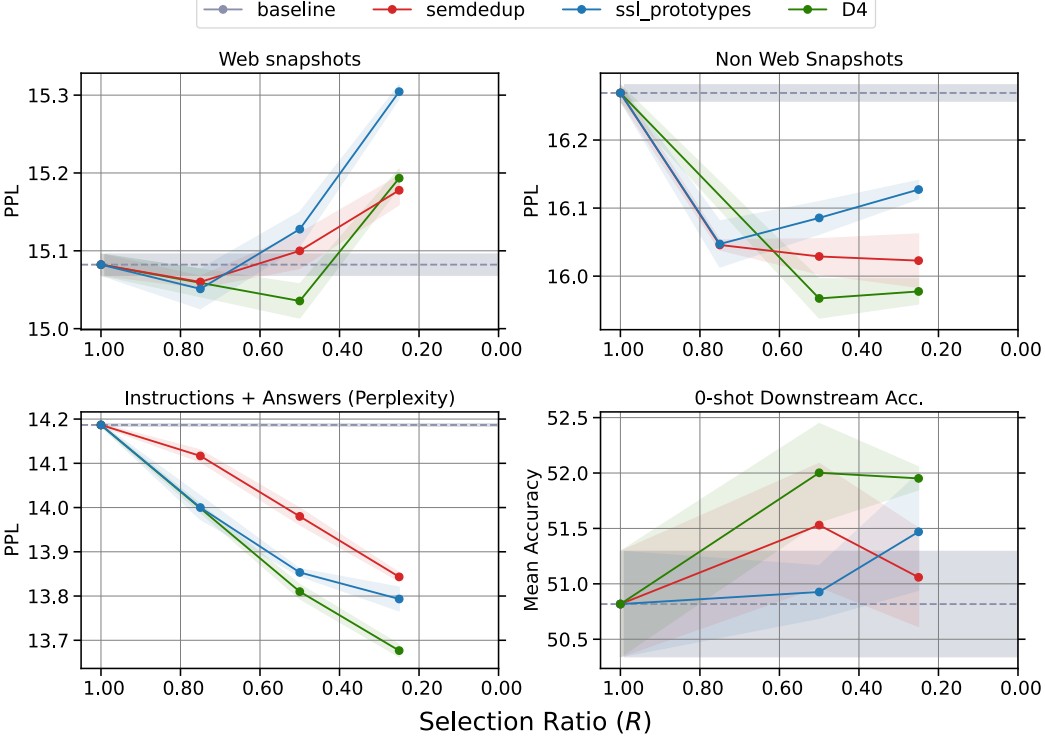

Figure 2: Comparison of data selection methods on validation perplexity. Each point denotes a 1.3B OPT model trained on 40B tokens. The x-axis denotes the selection ratio $R$. The y-axis for the top 2 and bottom left graph depicts perplexity; the bottom right graph is average downstream on 16 NLP tasks from Zhang et al. [59]. The grey line denotes the value for baseline training. Shaded error is standard error across 3 seeds. **Each point on this graph is trained on the same token budget**: when we decrease $R$, we jointly increase the size of the source dataset (e.g. choosing 1/4 of documents from a 4x'ed sized source dataset).

## 4.1  Fixed compute regime: can data selection help on fixed token budgets?

In this section, we consider the fixed compute setting, where we curate and train on a fixed token budget by jointly increasing the size of the source dataset $D_{source}$ and decreasing $R$ (the fraction of the $D_{source}$ which is selected), such that the target token budget remains constant. This setting is analogous to the most common paradigm for LLM training. As $D_{source}$ grows and $R$ decreases, we select from larger and larger initial datasets, resulting in a larger set of high-quality data points to select from and increasing the overall quality of the selected set. For clarity, we plot performance as a function of the ratio of the $D_{source}$ to $D_{target}$. For each setting, we evaluate the performance of a baseline, SemDeDup alone, SSL Prototypes alone, and our proposed method D4.

**Validation Perplexity.** In Figure 2, we show that a relatively small amount of data selection using any of the three methods (small $R$) brings consistent improvements on all validation sets. However, as we increase $R$, we observe *opposing effects* on web snapshot and non-web-snapshots validation sets. We analyze this discrepancy in-depth in Section 4.4. However, on the Instruct OPT validation set, which corresponds much more closely to the the high-quality generations we want our LLMs to achieve, we found that all three methods led to consistent and clear perplexity improvements. Notably, we found that while all three methods provided benefits, D4 outperformed using both SemDeDup and SSL Prototypes independently, with the most notable gains exhibited when the source dataset is around 4x the target dataset size. Given that D4 consistently improves with source dataset size, we estimate this gap to grow with source dataset size.

**Downstream Task Accuracy.** In Figure 2, we also report 0-shot downstream accuracy averaged across a suite of NLP tasks. While the high variance of downstream accuracy makes it challenging to identify clear trends in the performance of various models, we again observe that 0-shot downstream accuracy generally increases with source dataset size.

Our findings also hold at larger model scales. We pick our best-performing configuration from 1.3B OPT experiments (e.g., $R = 0.25$) and train 6.7B OPT models on 100B tokens. Figure 1 shows the positive effects of applying D4 with $R = 0.25$ for a 6.7B model. The model trained on the pruned data reaches the same perplexity as the baseline model using 20% fewer update steps on average and achieves a 2% improvement in accuracy on our suite of downstream tasks at the end of the training - about as much difference as was reported by Zhang et al. [59] between the OPT and GPT-3 family of models on the same set of tasks (See Figure 3 of Zhang et al. [59]).

## 4.2  Fixed data regime: what happens when we run out of data?

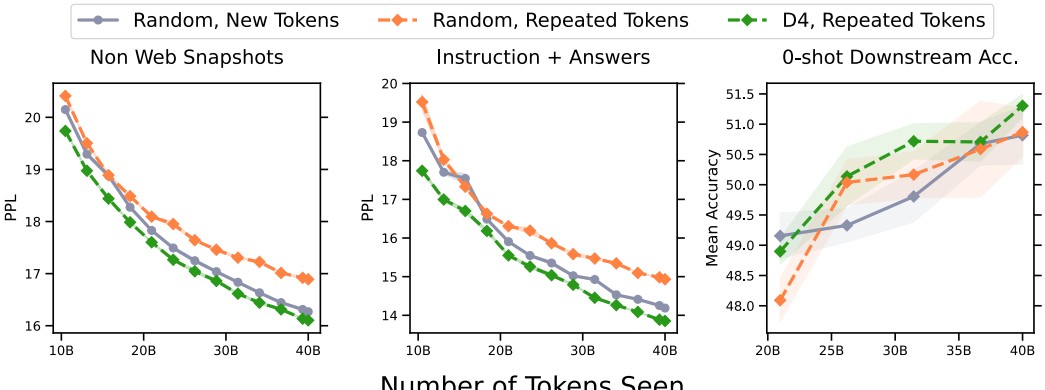

Figure 3: Comparing new tokens vs. repeated tokens for random data selection and D4 for fixed selection ratio $R = 0.25$ for 1.3B OPT pre-training. Each method chooses 25% of documents from the source dataset $D_{source}$, and epochs over that subset until the target token budget of 40B is reached. We observe that repeating tokens via D4 outperforms baseline training (random, new tokens).

The results in Section 4.1 indicate that, given a fixed amount of compute for training, selecting data from larger and larger source datasets is a promising method to improve language model performance. However, there is a practical limit to how much data can be curated from the web and, therefore, a

| $S$ | $T_{total}$ | $T_{selected}$ | Epochs | Non-Web Snapshot PPL | Instruction + Answers PPL |
|---|---|---|---|---|---|
| Random | 40B | 40B | 1 | $16.27 \pm 0.012$ | $14.19 \pm 0.003$ |
| | 40B | 20B | 2 | $16.39 \pm 0.011$ (+0.12) | $14.37 \pm 0.015$ (+0.18) |
| D4 | 40B | 20B | 2 | **16.10** $\pm 0.024$ (-0.17) | **13.85** $\pm 0.016$ (−0.34) |

Table 1: For fixed data selection method and source dataset size, we compare the effects of choosing new tokens or repeating token. All models are 1.3B OPT models trained on 40B tokens. $T_{selected}$ denotes the number of tokens selected from the source dataset. The top row denotes baseline training. Mean and standard error across 3 seeds are shown. **Surprisingly, cleverly choosing tokens to repeat via D4 outperforms randomly selecting new tokens.**

natural limit to the size of the source dataset. What happens when we run out of data? Hernandez et al. [19] found and analyzed disproportionately adverse effects of repeated data points in the training data. Similarly, concurrently to our work Muennighoff et al. [37] shows that test loss deteriorates when epoching over a random subset of C4 more than four times. In this section, we investigate how the use of D4 affects model performance in this limited data, multi-epoch setting.

To test this, we assume a fixed token budget and a fixed data size which matches the token budget. We evaluate training on all the data as well as for two epochs on subsets of the data selected either randomly or using D4. We trained 1.3B parameter OPT models on these configurations and report average perplexity in Table 1. Unsurprisingly, epoching over a randomly selected subset of the data instead of using all the available data once leads to a slight degradation in model perplexity. In contrast, repeating data selected by D4 leads to an improvement in perplexity and downstream accuracy over randomly sampling new tokens. In other words, it is beneficial to select data via D4 and epoch 2 times, instead of doing one-pass learning on all available data. As seen in Figure 3, this finding generally holds across training as well. We refer to Section A.6 for results across model scale and data selection ratio.

To the best of our knowledge, this is the first result to demonstrate the benefits of repeating data for LLMs over randomly sampling new tokens via a principled data selection technique. We argue that the optimal way of using large-scale web data to pre-train LLMs could be: strategically choose a significantly smaller but better-distributed subset of the data and epoch over it multiple times.

### 4.3 Cost of data selection

In Section 4.1, we find that by training a 6.7B parameter model on data selected by D4, we reach the final perplexity of a baseline model using 20% fewer model updates. In our particular setup, this translates to **saving approximately 4300 GPU hours** - we will refer to this as the *naive* efficiency gain as it does not account for the the cost of computing the selection metric.

To demonstrate our method's practicality, we must ensure the cost of selecting data is significantly less than this. As described in Section 3.4, selecting data via D4 involves: first, embedding documents via a 125M OPT model; second, computing K-Means indices + distance to indices. The first step is completed on a single machine with 96 CPU cores in approximately one day. Given the two orders of magnitude difference between the prices of CPU and GPU cores [1], we consider this cost negligible. For the second step, embedding 400B tokens with a 125M parameter model takes approximately 888 GPU hours, using the same A100 GPUs. Subtracting this from the *naive* efficiency gain of 4300 GPU hours, we arrive at an *overall* efficiency gain of 3412 GPU hours. This is how much compute D4 saved us in practice when training our single 6.7B parameter model. In Figure 4, we redo this calculation for different model sizes and we see that *overall* efficiency gain

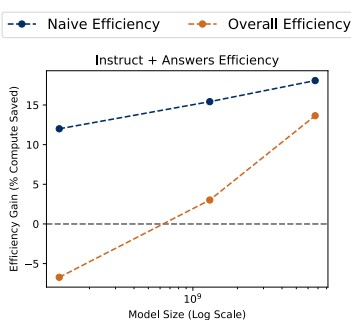

Figure 4: *Naive* and *overall* efficiency gain of data selection via D4 relative to the total cost of training as a function of model size on Instruct + Answers perplexity at $R = 0.25$.

[1]Source: https://aws.amazon.com/ec2/pricing/on-demand/

increases with model size. Based on this, we can conservatively estimate that D4 would have overall efficiency gains of 20% for LLama-65B [50] and 22% for OPT-175B [59].

## 4.4 Analysis of D4

### 4.4.1 Why does data selection hurt performance on web snapshots?

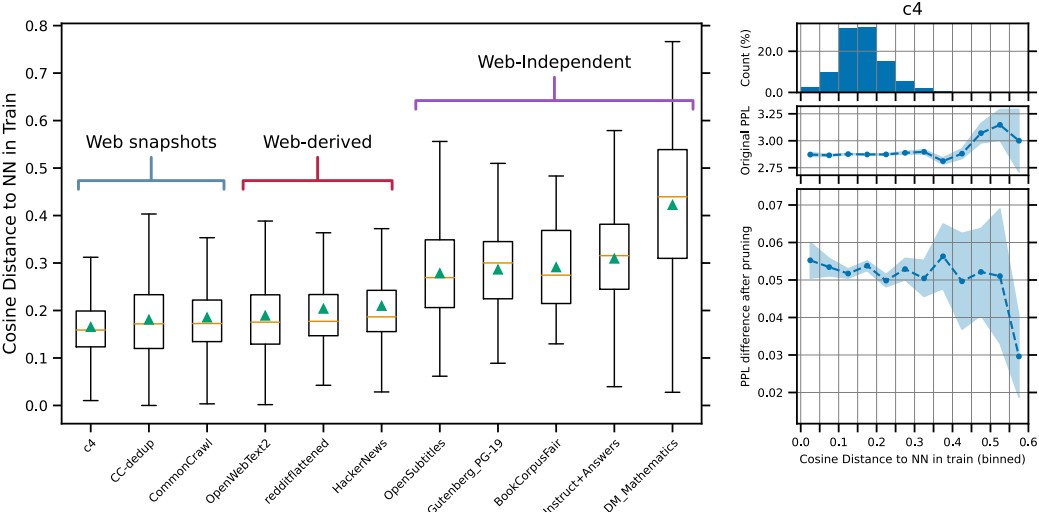

Figure 5: **Left**: Train-test similarity across validation sets. X-axis denotes the name of the validation set (refer to Section 3.4 for more information about each validation set), and y-axis denotes the cosine distance to the nearest neighbor in the training set for the 1.3B OPT 40B baseline (the green triangle denotes mean, and the yellow bar denotes median). We observe that web-snapshots validation sets are closest to points in the training set and are disproportionately affected by data selection. **Right**: Analysis of C4 validation set. (Top): Histogram of cosine distance to nearest neighbor in train. For each bin, we show the mean original perplexity (middle) and mean difference in perplexity after data selection (bottom). "Easy" (low original ppl) points close to the training set are generally the points most affected by data selection.

While we observe consistent *average* perplexity improvements, Section A.3 demonstrates that this perplexity improvement varies greatly across validation sets. More importantly, data selection always impairs performance on web snapshot validation sets such as CC-dedup, CommonCrawl, and C4. To investigate why this occurs, we embed each validation set into the same embedding space as the training set and search for the nearest neighbors to validation points in the training set for our 1.3B baseline model. In the left plot of Figure 5, we show that validation sets drawn from the same distribution as web-snapshots are substantially closer to training set compared to other validation sets. The right plot of Figure 5 shows that data selection disproportionately affects web-snapshot validation sets. In the top-right plot, we see that web validation sets reside in regions of the embedding space which are sparsified as a result of data selection (e.g. regions of space close to cluster centroids in the training set), and in the bottom-right plot we see that these points are also the most affected by data selection, since their perplexity after data selection significantly increases. Moreover, the middle-right plot shows that these validation points have the lowest perplexity before pruning indicating that these points are "easy" points, perhaps due to their proximity to the training set.

Given that some of our validation sets are extremely close to the training set, we question whether they are still strong indicators of generalization. In fact, in Figure 6, we find evidence of a slight inverse relationship between perplexity on web snapshots and more robust indicators of LM ability, such as perplexity on instruction-tuned datasets and downstream accuracy. In contrast, we observe that perplexity on Instruct+Answers is positively correlated with downstream accuracy, suggesting that validation perplexity on instruction tuned data is a better measure of model quality. For this reason, we group most of our results in Section 4 into Web Snapshots and Non-web Snapshots (which consists of Web-Derived + Web-Independent from Figure 5, see Section A.1.4 for a full-list of validation set names).

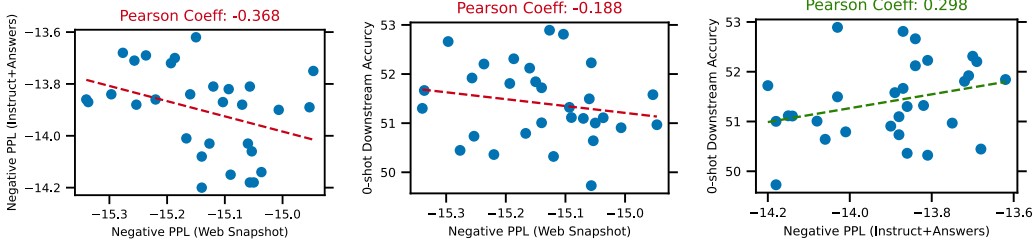

Figure 6: Correlation between (left): negative Instruct+Answers perplexity and negative web snapshot perplexity, (middle): Downstream accuracy and negative web snapshot perplexity, (right): Downstream accuracy and negative Instruct+Answers perplexity. Each point is one training configuration (1.3B OPT model, 40B tokens), with the only change being the data selection method and pretraining seed. Web snapshot perplexity is slightly negatively correlated with stronger indicators of LM ability.

### 4.4.2 Importance of re-clustering between SemDeDup and SSL Prototypes

As mentioned in Section 3.4, we hypothesize that sparsifying dense regions of space containing excessive semantic duplicates improves the clustering quality and is, therefore, critical to the performance of D4. To isolate the effect of re-clustering on D4, we run experiments with a version of D4 where we remove the re-clustering step (e.g. we keep the original clustering). As shown in Figure 7, omitting the re-clustering step significantly worsens performance, and we observe in the rightmost plot of Figure 7 that SemDeDup indeed removes extremely dense clusters surrounding centroids (e.g. duplicate-driven clusters). We analyze this in more depth in Section A.9.

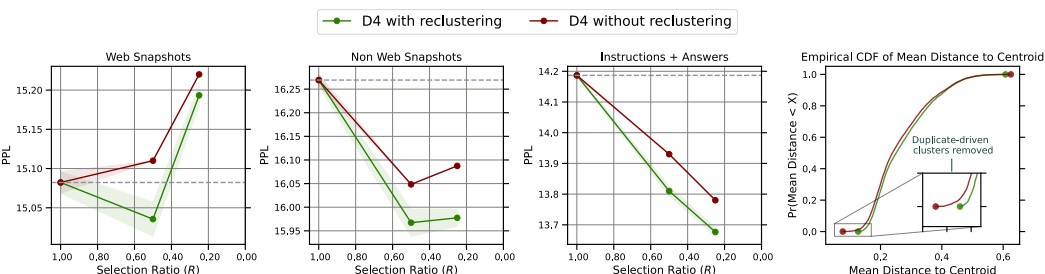

Figure 7: Investigating the necessity of the re-clustering step in D4. We see that re-clustering improves perplexity across Web snapshots (left), Non-web snapshots (middle-left), and Instruct + Answers (middle-right). Right: Empirical CDF of mean distance to centroid, with and without re-clustering. Re-clustering removes duplicate driven clusters (clusters with low mean distance to centroid).

## 5 Summary and Limitations

We introduced D4, a method for data curation on LLMs that improves training efficiency by ~20% across multiple model scales, with larger gains at increased model scale. We also demonstrated that, in contrast to common practice, repeating data via epoching can be beneficial for LLM training, but only if the data subset is intelligently selected. While we have shown encouraging efficiency gains and performance improvements via D4, our work has several limitations and many future directions.

**Mixing different training distributions:** While we chose one data distribution to both select data and train on, modern LLM setups usually mix different data sources. Our method is likely complimentary to such pipelines: practitioners may use D4 to diversify and de-duplicate individual data sources and then mix data sources to provide additional diversity in their training dataset. We leave exploring the efficacy of D4 on a mix of training distributions as future work, but expect that this will yield further gains by reducing redundancy across datasets as well as within datasets.

**Model scale:** Due to compute limitations, the largest models we evaluated were 6.7B parameters trained on 100B tokens. While, to our knowledge, this is the largest to date application of embedding based data curation approaches, further investigation at model scales exceeding 100B would be very interesting, particularly in light of our observation that the efficiency gain grows with model scale.

# 6  Acknowledgements

The authors would like to thank many people who helped bring this work to fruition: Srini Iyer, Yuchen Zhang, Todor Mihaylov, Jacob Xu Moya Chen, Mansheej Paul, Mitchell Wortsman, Amro Abbas, Aaditya Singh, Myra Cheng, and Matthew Leavitt. The authors would also like to thank Surya Ganguli, Mona Diab, and Xian Li for initial brainstorming and are grateful for help with compute infrastructure given by Henry Estela and Victoria Lin. Lastly, the authors would like to thank anonymous reviewers for improving the quality and writing of this paper.

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
