# A  Appendix

## A.1  Experimental Setup Details

### A.1.1  Hyperparameters for model training

As mentioned in Section 3.4, we use the same hyperparameters and configurations as the original OPT model architecture from Zhang et al. [59]. We describe these hyperparameters briefly in Table A1. We chose these configurations because they are openly available and have been used as the standard in many previous works [1, 13, 29, 48, 59]. All models use GELU activation [18], Adam optimizer [26] with $\beta_1 = 0.9$, $\beta_2 = 0.95$, $\epsilon = 10^{-8}$, weight decay set to 0.1, and we clip gradient norms at 1.0. We use a polynomial learning rate schedule, where learning rate warms up from 0.0 to peak learning rate over the first 375 million tokens, and is then annealed to (0.1 * Peak LR) over the remaining $(T_{target} - 375)$ M tokens. We train all our models in fully sharded data parallel mode [2] using Megatron-LM Tensor Parallelism [45] with fp16 precision. For reproducibility (and perhaps the only difference from the original configuration in Zhang et al. [59]) is that we do not use dropout.

Table A1: Model architecture details. Most of the parameter configurations are the same as in Table 1 of Zhang et al. [59]. Batch size denotes the total tokens that the model sees during one gradient descent update.

| Scale | Num Layers | Num Heads | Embedding Dim | Peak Learning Rate (LR) | Batch Size |
|-------|-----------|-----------|---------------|-------------------------|------------|
| 8M    | 4         | 2         | 128           | 1.0e-3                  | 0.5M       |
| 125M  | 12        | 12        | 768           | 6.0e-4                  | 0.5M       |
| 1.3B  | 24        | 32        | 2048          | 2.0e-4                  | 1M         |
| 6.7B  | 32        | 32        | 4096          | 1.2e-4                  | 2M         |

### A.1.2  Dataset Curation Details

In this subsection, we describe how we curate *CC-dedup*, the starting source dataset used throughout the paper. We start with 5 CommonCrawl dumps [2] which range from 2017 to 2020. We then use CC-net [54], to de-duplicate data at the paragraph level, remove non-English web pages, and filter out low-quality pages. The pipeline we use is identical to the pipeline used in Touvron et al. [50] (see the section after the subtitle "English CommonCrawl [67%]", within Section 2).

On top of this, we add an additional step of MinHash [8] de-duplication at the document-level. The parameters for MinHash are 20 hashes per signature, 20 buckets, and 1 row per bucket. These parameters are the default parameters in the spark implementation of MinHashLSH, and we did not do a hyperparameter sweep on these parameters due to compute limitations. Previous work has attempted running MinHash with much more aggressive parameters: Lee et al. [27] and Penedo et al. [39] use 20 buckets, 450 hashes per bucket, and 9000 signatures per hash. We conjecture that more aggressive MinHash would remove more templates, resulting in a higher-quality starting dataset, potentially making the SemDeDup step of D4 less necessary. Abbas et al. [1] did find that the performance of MinHash from Lee et al. [27] and SemDeDup are comparable at a fixed data selection ratio of 3.9% on C4, indicating that SemDeDup filters out similar data to aggressive MinHash does. We leave sweeping over these hyperparameters as future work.

We note that since our dataset is curated from CommonCrawl dumps, there is risk that our training set contains offensive or PII content. We note, however, that this risk is no more than that of standard language modeling curation such as Touvron et al. [50], since we use the same pipeline to filter CommonCrawl dumps.

### A.1.3  Parameters for Data Selection

All methods introduced in Section 3.4 involve clustering embeddings using K-Means. Our starting training dataset CC-dedup contains roughly 600 million documents in total. Running K-Means clustering on all 600 million 768-sized vectors would take a considerable amount of compute. Instead, we follow previous work [1, 47] and randomly sample roughly 100M documents with which to

---

[2] https://commoncrawl.org/the-data/get-started/

calculate centroids. We normalize the embeddings for these 100M documents to have L2-norm of 1.0, and then use faiss [24] with the following parameters:

```
faiss.Kmeans(
    768 # 125M OPT model embedding size,
    11000 # 11K clusters,
    niter=20 # 20 iterations,
    verbose=True,
    seed=0,
    gpu=False,
    spherical=True,
    min_points_per_centroid=1,
    max_points_per_centroid=100000000
)
```

We choose 11000 clusters following previous work [1] and we note that this choice sticks to the heuristic that the number of clusters should roughly be the square root of the number of total points being clustered. We also note that in initial experiments for data selection at the 125M OPT model scale, we did not find a significant effect of number of clusters on the performance of our data selection methods (see Figure A1) this finding agrees with Abbas et al. [1] who notice significant overlap between datasets selected by SemDeDup with different number of clusters (see Figure A2 in Abbas et al. [1]).

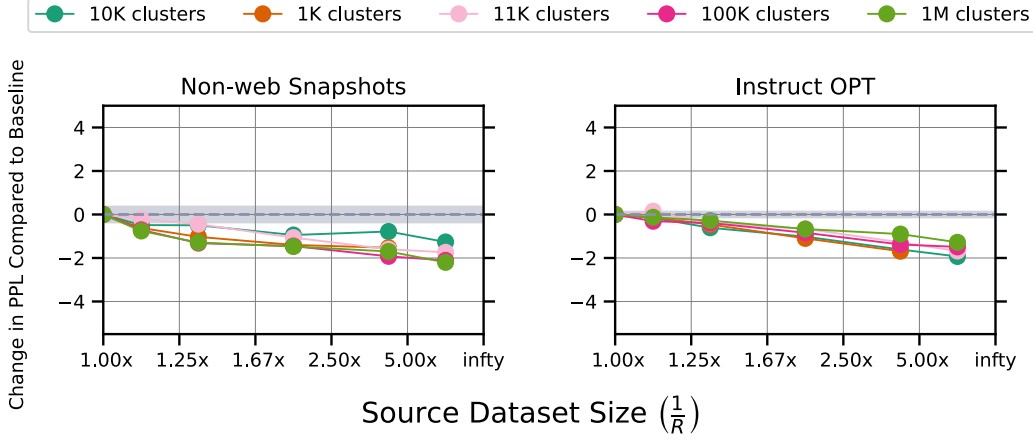

Figure A1: Effect of number of clusters in K-Means on data selection performance. All models are 125M OPT models, where the training set (and starting source dataset) is C4 and we select data with SSL prototypes. The y-axis is the change in perplexity compared to baseline training, meaning that baseline training is at 0.0, and going *down* on the graphs indicates *better* performance. The x-axis is the source dataset size. We show results for average perplexity on Non-web snapshot validation sets (left) and Instruct + Answers (right). We notice that there is not a significant difference when changing number of clusters (e.g. if we drew error bars around each line, they would all be overlapping), but 11K clusters is generally among the top-3 best performing methods.

We deliberately set min points per centroids low and max points per centroid high so that faiss does not attempt to manually balance the clusters while doing K-Means. Sorscher et al. [47] found that explicitly class-balancing is important: they introduce the "class balance score" (see Section H of Sorscher et al. [47]) which is the expectation of the quantity $\frac{\text{size of majority class}}{\text{size of minority class}}$ over all pairs of classes. They then set a hard limit for the class balance score of 0.5, meaning that "every class has at least 50% of the images that it would have when pruning all classes equally" [47]. We consider the unsupervised-learning analog of the class-balance score, which we refer to as the "cluster balance" score. The cluster balance score is the expectation of the quantity $\frac{\text{size of bigger cluster}}{\text{size of smaller cluster}}$ over all pairs of clusters. Across all of our data selection methods (and choices for R) we find that this value is generally equal to or bigger than 0.5 without any explicit intervention. For this reason, we do not

explicitly cluster balance, although we note that changing how many points are sampled from each cluster (based on properties of the cluster) is very interesting future work.

D4 parameters: The choice of parameters $R_{proto}$ and $R_{dedup}$ while using D4 will have impact on the performance of D4. Given limited compute, we are not able to sweep over these hyperparameters. Instead, we strategically choose these parameters: we first look at the highest value of $R$ in SemDeDup that results in perplexity improvement across validation sets. We choose the "highest value" because the purpose of SemDeDup is to remove duplicate-driven clusters and low $R$ with SemDeDup generally removes more than just templates/semantic duplicates. As seen in Section A.3, this generally occured with $R_{dedup} = 0.75$. Thus, we chose $R_{dedup} = 0.75$ and varied $R_{proto}$ to obtain different data selection ratios for D4.

### A.1.4 Which validation sets go into the averages?

For clarity, we explicitly state the validation sets which we consider "Web Snapshots", "Non Web Snapshots", and "Instruct + Answers" when reporting averages:

**Web Snapshots**: perplexity on validation set of C4, CC-dedup, CommonCrawl (from the Pile)

**Non-web Snapshots**: perplexity other validation sets from the Pile, comprising of OpenWebText2, HackerNews, Wikipedia (en), BookCorpusFair, DM Mathematics, Gutenberg PG-19, OpenSubtitles, and USPTO. Also included in this average is "redditflattened" (validation set from Pusshift.io Reddit [4]), "stories", "prompts_with_answers" (which is described below) and "prompts" (which is the same as "prompts_with_answers" but where each sample is just the instruction-tuning prompt without the answer).

**Instruct + Answers**: perplexity on instruction-tuning data from OPT-IML [21], where each sample contains both the instruction-tuning prompt and the answer (in Figure A4 this is referred to as "prompts_with_answers."

While the validation sets in web-snapshots and non-web snapshots are clear (they are either standard open-sourced datasets, or derived from commonly used data), we expect that the "Instruct + Answers" data might be new to some readers. We provide a few examples of what this validation set looks like in Table A2.

Table A2: Examples from "Instruct + Answers" validation set

| Raw Text |
|---|
| Instructions: In this task, you are given two phrases: Head and Tail, separated with <sep>. The Head and the Tail events are short phrases possibly involving participants. The names of specific people have been replaced by generic words (e.g., PersonX, PersonY, PersonZ). PersonX is always the subject of the event. You have to determine whether the Head is located or can be found at/in/on the Tail or not. Classify your answers into "Yes" and "No". The phrase may also contain "___", a placeholder that can be an object, a person, and/or an action.Input: Head: PersonX acknowledges gratefully the ___<sep>Tail: to use it Output: No |
| Read the given sentence and if it is a general advice then indicate via "yes". Otherwise indicate via "no". advice is basically offering suggestions about the best course of action to someone. advice can come in a variety of forms, for example Direct advice and Indirect advice. (1) Direct advice: Using words (e.g., suggest, advice, recommend), verbs (e.g., can, could, should, may), or using questions (e.g., why don't you's, how about, have you thought about). (2) Indirect advice: contains hints from personal experiences with the intention for someone to do the same thing or statements that imply an action should (or should not) be taken. Input: Let it go. Output: yes" |
| Instructions: You are given a sentence in English. Your job is to translate the English sentence into Italian. No! Demand to understand. Ask. Answer: No! Esigete di comprendere. Chiedete. |
| Task: In this task you will be given a list of integers. You should round each integer to the nearest tens place. That means you should round the number to the nearest multiple of 10.Input: [528, -636, -686, 368, -433, 992, 886] Answer: [530, -640, -690, 370, -430, 990, 890] |

## A.2   Efficiency gains across model scales and training

In this section, we investigate the relationship between model scale, and performance gain obtained by selecting data via D4. Specifically, we train three groups of models: 125M OPT models trained on $T_{target} = 3B$ tokens, 1.3B OPT models trained on $T_{target} = 40B$ tokens, and 6.7B OPT models trained on $T_{target} = 100B$ tokens. We notice in Figure A2 that D4 results in efficiency gains across the board in terms of perplexity. Surprisingly, these efficiency gains seem to increase with scale, indicating that at bigger model scales, D4 might lead to even more efficiency gains. We also see efficiency gains in 0-shot downstream accuracy for 1.3B and 6.7B model scales on the order of 30% for both 1.3B and 6.7B models, but we note that evaluation downstream performance on intermediate checkpoints is not completely fair due to unfinished learning rate schedule. Nonetheless, we see that downstream accuracy efficiency gains are not decreasing with scale.

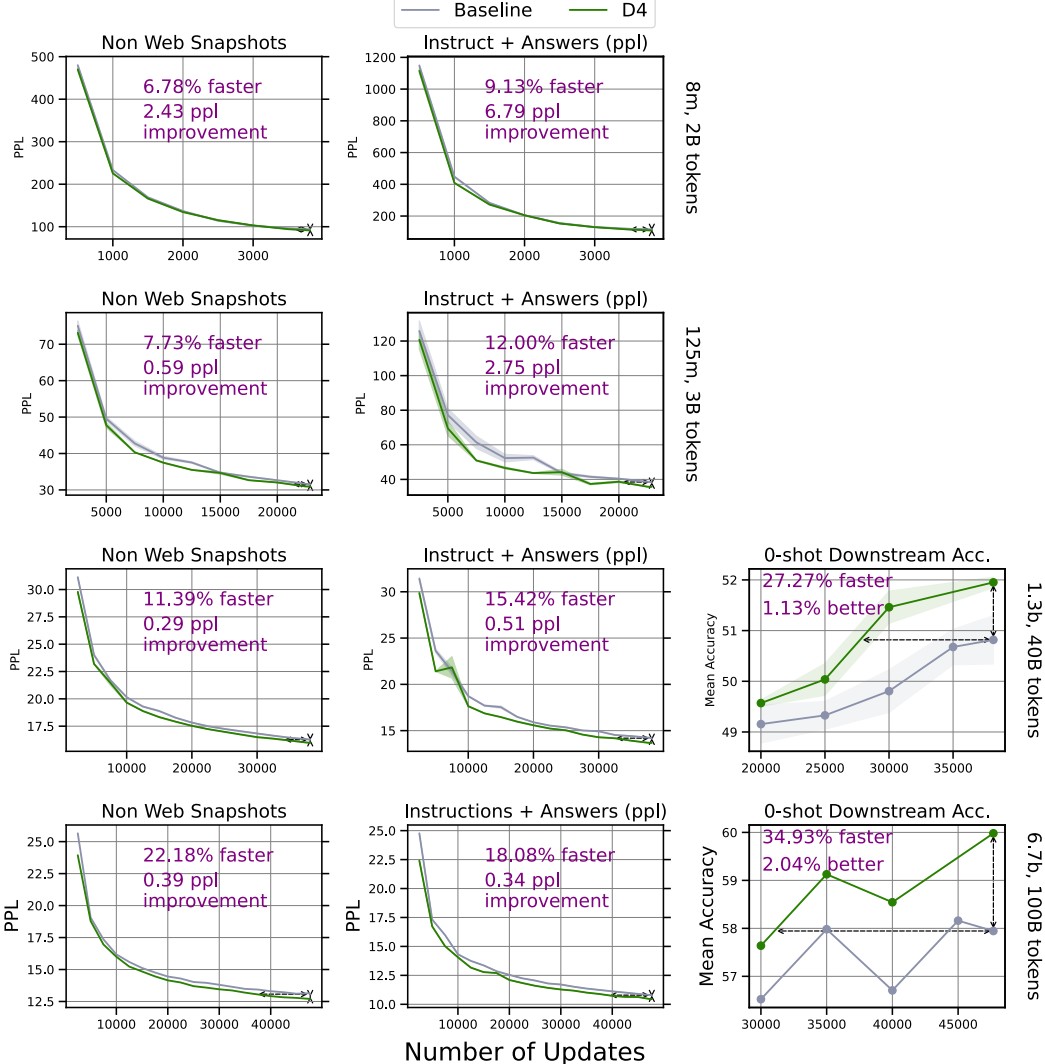

Figure A2: Training trajectory of OPT models trained on raw data (gray line) and data selected via D4 (pink line). Across model scales (1st row: 8M OPT models trained on 2B tokens, 2nd row: 125M OPT models trained on 3B tokens, 3rd row: 1.3B OPT models trained on 40B tokens, 4th row: 6.7B OPT models trained on 100B tokens), we see significant efficiency gains in both perplexity (left two columns) and 0-shot downstream accuracy on 16 NLP tasks (right column). Importantly, we see that increasing model scale does not decrease efficiency gains. All plots show mean and standard error across three seeds, except for the last row. We do not evaluate downstream accuracy for models smaller than 1.3B because they are likely too close to random performance to indicate whether a particular data selection method is better.

## A.3 Individual Breakdowns of Downstream Accuracy and PPL

In Section 4, we see that D4, SSL prototypes, and SemDeDup achieves significant gains on perplexity (averaged across different validation sets) and downstream accuracy (averaged across different NLP tasks) compared to baseline training. Further, we generally see that D4 outperforms SSL prototypes and SemDeDup. In this section, we provide a more fine-grained analysis of these claims across individual tasks.

For perplexity, we notice in Figure A4 that the claims in Section 4 generally hold across validation sets: for web snapshots validation sets such C4, CC-dedup, and CommonCrawl, we see performance

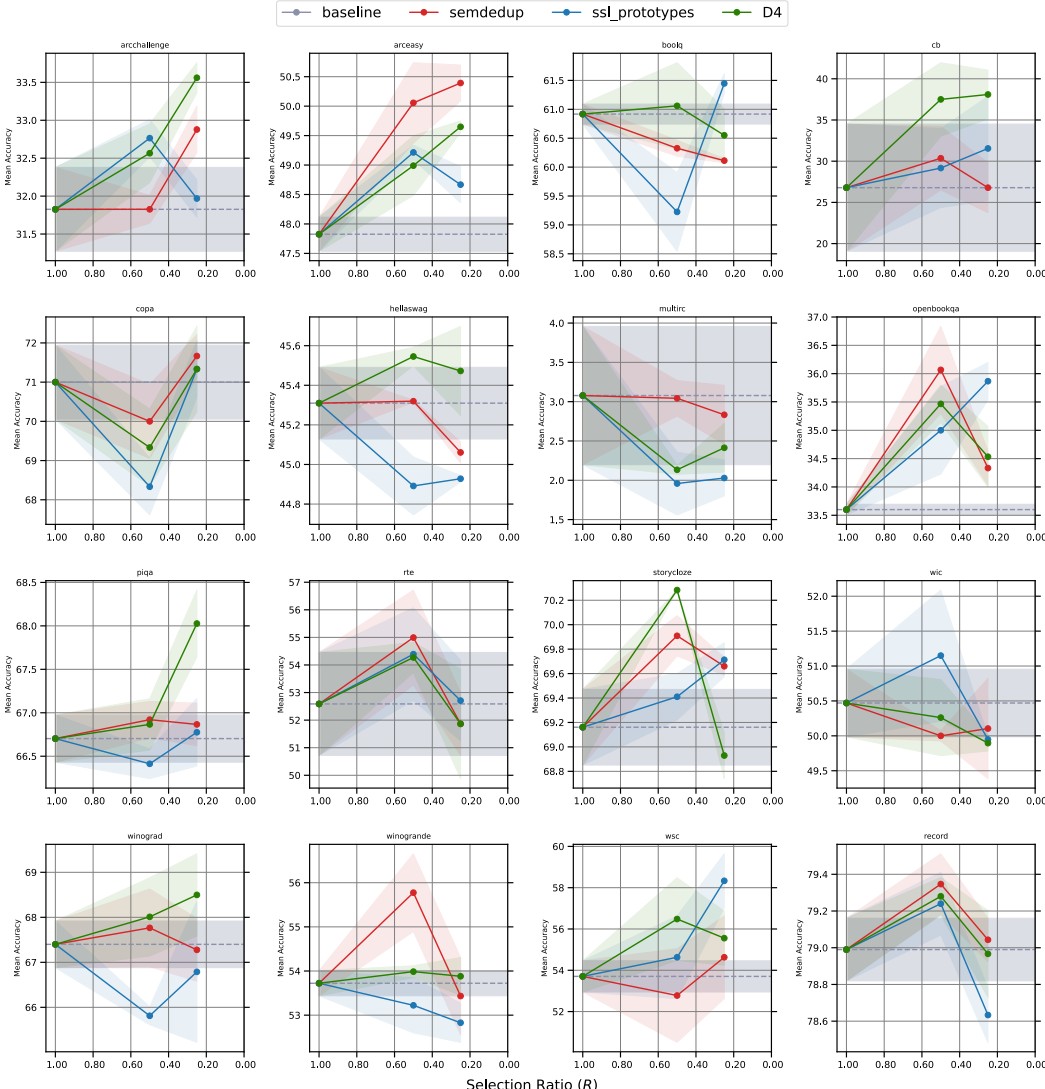

Figure A3: Per-task breakdown of 0-shot downstream accuracy comparison across data selection methods, for 1.3B, 40B OPT model. For a description of the 16 NLP tasks shown above, see Section 3.4. We note that there is considerable variability across individual downstream tasks.

worsens with data selection compared to baseline training, and that D4 generally has the slowest rate of performance degradation. We note that, across all non web-snapshot validation sets, there is no clear winner among data selection methods. We emphasize however that *we observe consistent improvement over baseline training on most validation sets* we use — for example in Figure A4 we observe that, when selecting tokens from a 1.25x source dataset, all data selection methods improve over baseline across all validation sets except C4 and CC-dedup (however, as we explain in Section 4.4, this decrease in performance on C4 and CC-dedup is expected).

For downstream accuracy, we chose to match the exact downstream evaluation done in Zhang et al. [59] since we use OPT architecture and hyperparameters. Similar to Zhang et al. [59], we notice considerable variability across the 16 NLP tasks in Figure A3, motivating us to look at the mean downstream accuracy across tasks.

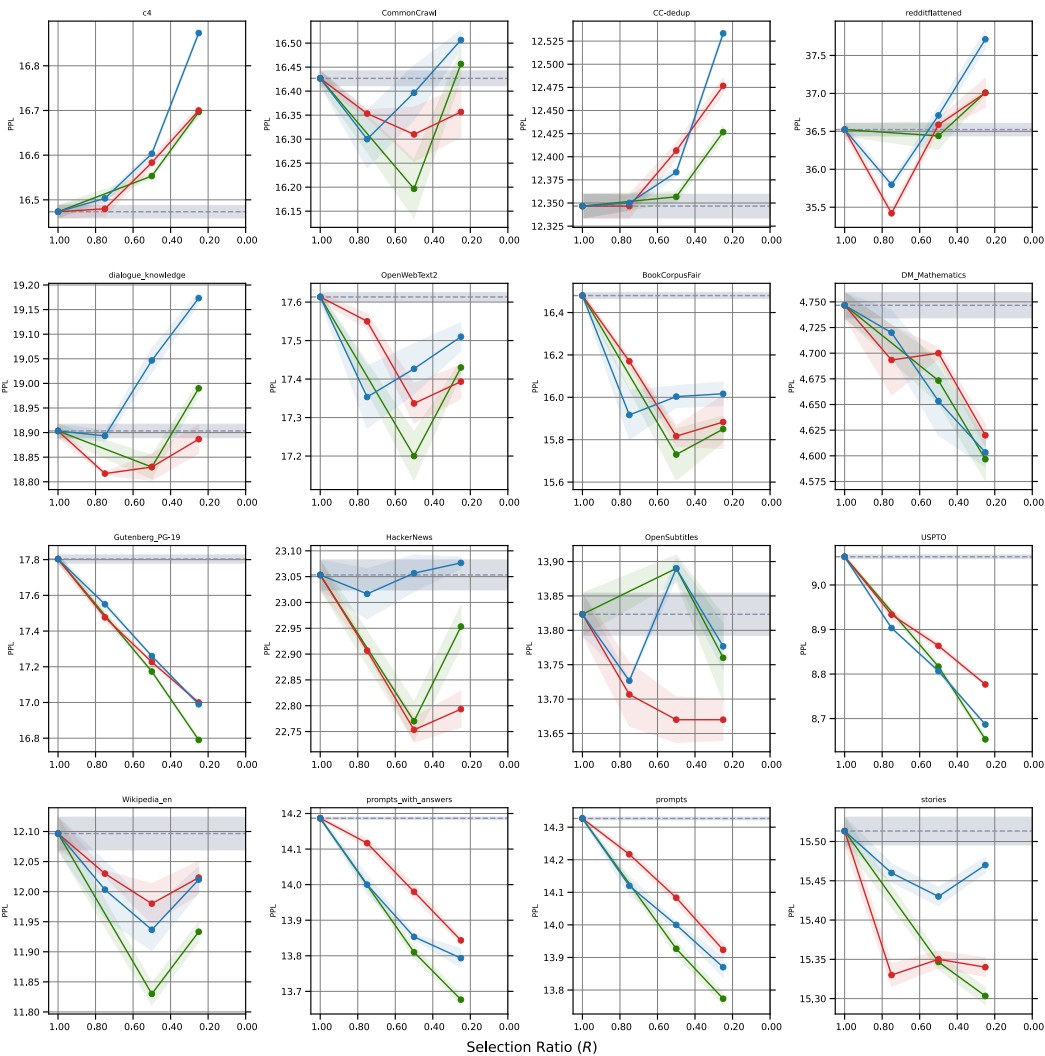

Figure A4: Perplexity as a function of source dataset size for 1.3B OPT model 40B token training runs, across data selection runs. Each plot above represents perplexity on an individual validation set (see Section 3.4 for more information). Mean and standard error across 3 seeds is shown (standard error is denoted by shaded regions).

## A.4 SSL prototypes and SemDeDup overlap

Figure A5 shows the overlap between datasets selected by SemDeDup and SSL Prototypes. While the two methods do not arrive at the same set of data points, there is a significant overlap between the datasets curated by the two methods. We hypothesize that this is because both SSL prototypes and SemDeDup prune away dense regions of space surrounding cluster centroids: by definition, SemDeDup sparsifies dense regions of space within a cluster; similarly, by definition, SSL prototypes will prune away datapoints close to the cluster centroids. Since K-means clustering places centroids in dense regions of space (see Figure A6 where we observe that the distribution of cosine distances to cluster centroid is skewed right), we know that the regions of space surroundings centroids will be dense, and expect SSL prototypes and SemDedup to have significant overlap. Qualitatively, we inspect a few examples of points close to cluster centroids in Figure A3, Figure A4, Figure A5, and see that examples close to cluster centroids can be semantically redundant (e.g. templates). Therefore, it makes sense that any reasonable data selection strategy would prioritize sparsifying these dense regions of space surrounding cluster centroids. As mentioned in Section 3.4, sparsifying these dense regions of space containing excessive semantic duplicates is the original motivation behind D4. As

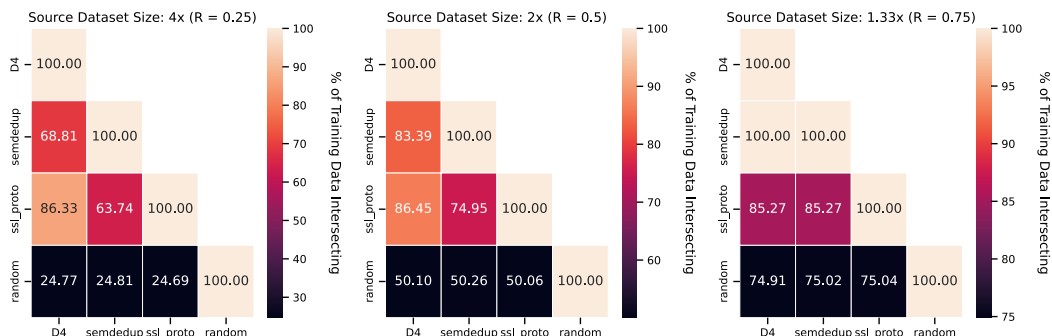

Figure A5: Similarity between data selection methods. Each square represents the percentage of training data that is intersecting, when selecting data via two different strategies. The $x$ and $y$ axis enumerate different data selection strategies.

shown in Figure 7, omitting the re-clustering step significantly worsens performance, and we observe in the rightmost plot of Figure 7 that SemDeDup indeed removes duplicate-driven clusters.

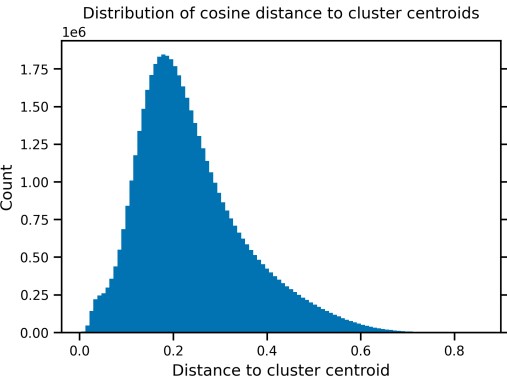

Figure A6: Distribution of cosine distance to cluster centroids for 50M randomly selected documents from the training set of CC-dedup. We notice that the distribution is skewed right, implying that datapoints are generally close to centroids.

## A.5  Investigating Train-Validation overlap

As briefly described in Section 4.4, we observe that many of our validation sets are close (in cosine distance) to our training sets, and the impact of data selection is varies across individual validation sets. Individual validation sets live in different regions of the embedding space, and as such they are affected differently by data selection. For example, one could imagine that web-snapshot validation sets such as C4 is close to CC-dedup in the embedding space, while esoteric validation sets (such as Gutenberg PG 19 or DM Mathematics) might be far. To quantify this, we first find the nearest neighbors in the training set to each validation point in all of our validation sets. We then qualitatively check (see Table A8 and Table A9 for examples) that nearest-neighbors in the training set truly convey information about validation points. we observe significant overlap between training points and validation points. We then quanitatively analyze how close each validation set is to the training set: in Figure A12, we show the breakdown of this distribution for each validation set. We see a general trend, that web-snapshots validation sets are closest to the training set as they are skewed to the right, while more esoteric validation sets (Gutenberg, or Wikipedia (en)) are more centered or even slightly left-skewed.

Motivated by this, we compare validation sets side-by-side (in terms of distance to training set) in Figure 5, and we see a similar trend. To further understand why different validation sets are affected differently by data selection, we loop through each data point in the validation set and record:

- distance to the training set e.g. how close is the validation point to the training set

- perplexity difference before and after data selection with D4 e.g. how much was this validation point affected by data selection

- original perplexity e.g. how easy was this data point originally

In Figure A11, we observe an interesting trend: for web-snapshot validation sets such as C4, the validation points closest to the training set are both (1) the easiest (lowest perplexity) points before data selection and (2) the points most affected by data selection. This seems to indicate that these validation points are "easy" due to their proximity to training points, and when these training points are removed from the training set due to data selection, the close-by validation points become difficult for the model. We do not see this trend on non-web snapshot validation sets such as DM Mathematics and Open Subtitles; in fact, we see an opposite trend where points furthest from the training set are generally most affected by data selection.

As a sanity check, we change the sizes of validation sets used to plot Figure 5 in Section 4.4. We see in Figure A8 that controlling for validation set size, we get the same jump going from web-derived to web-independent validation sets. In running this experiment, we are forced to randomly sample if the particular validation set is too big; to ensure that such random sampling does not change the distance to nearest neighbor in the training dataset too much, we vary the amount we sample for three differently sized datasets in Figure A7. We observe that changing the amount we randomly sample from a validation set does not significantly change the mean distance to nearest neighbor in train.

We also investigate whether the differences between validation sets in Figure 5 is due to training set size. We would expect that smaller training sets are "further" from validation sets, since (). Indeed we see this in Figure A9. However, we observe that the relative ordering of validation sets (with respect to average distance to the training set) remains the same for any fixed training dataset size. Moreover, we see in Figure A10 that the relative ranking of all validation sets as well as the jump from web-derived to web-independent validation sets from the original Figure 5 holds, even as we reduce training dataset size.

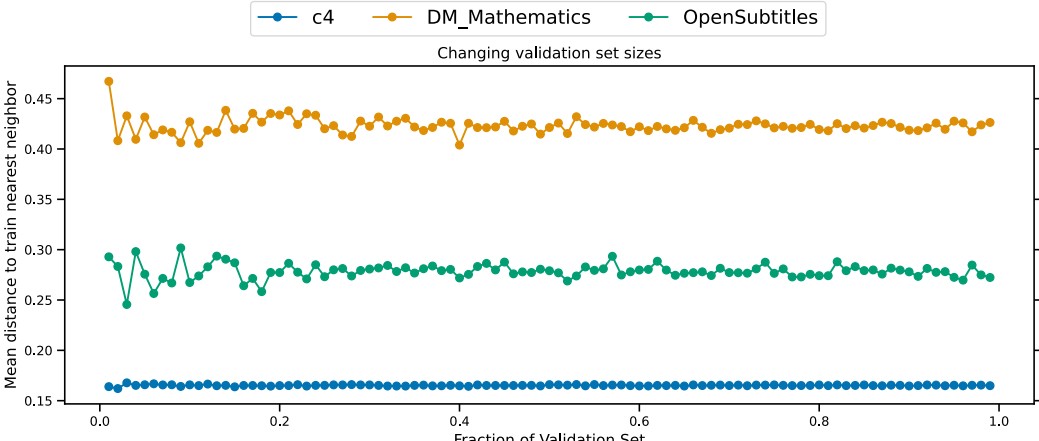

Figure A7: Studying the effect of validation set size on cosine distance to nearest-neighbor in training set. On the x-axis, we vary the size of the validation set (by randomly sampling the original larger validation set), and the y-axis represents distance to nearest neighbor in the training set (averaged across the validation set). We observe that regardless of what fraction of the original validation set is sampled, the mean distance to the nearest neighbor in train does not change, indicating that Figure 5 is not due to different validation set sizes.

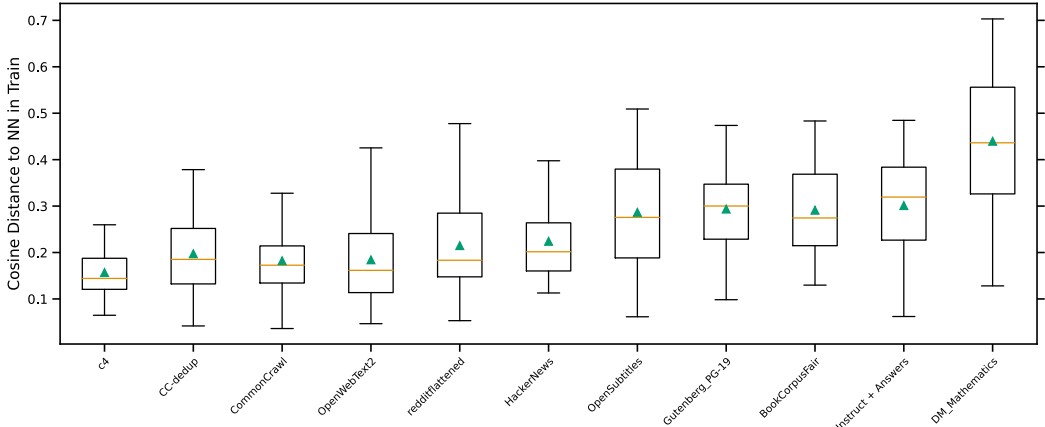

Figure A8: Investigating whether Figure 5 changes if we control for validation set size. In the Figure above, each validation set contains 50 data points, which is the size of the smallest validation set we use (BookCorpusFair). If a validation set is bigger than 50 data points, we randomly sample the validation set to obtain 50 data points.

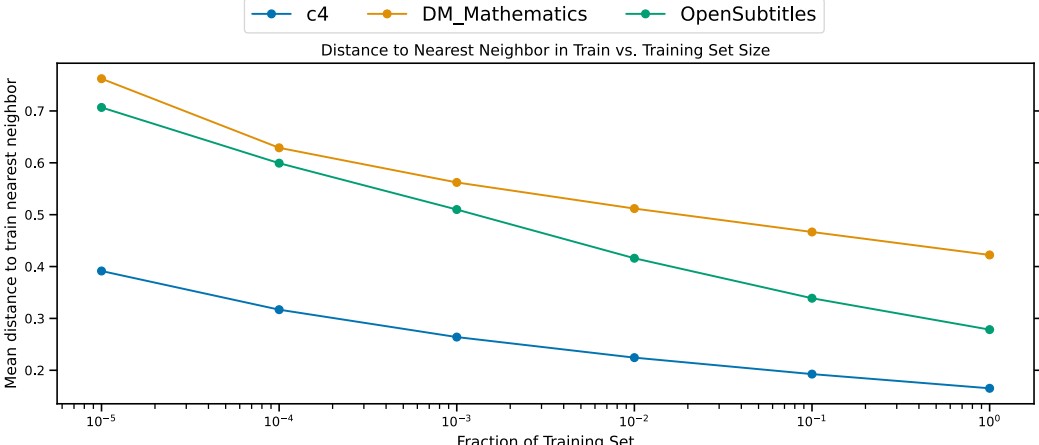

Figure A9: Studying the effect of training set set size on cosine distance to nearest-neighbor in training set. On the x-axis, we vary the size of the training set (by randomly sampling the original training set), and the y-axis represents distance to nearest neighbor in the training set (averaged across the validation set). We observe that cosine distance to the training set increases with smaller training sets, but the relative ordering of validation sets (with respect to mean distance to training set) remains the same.

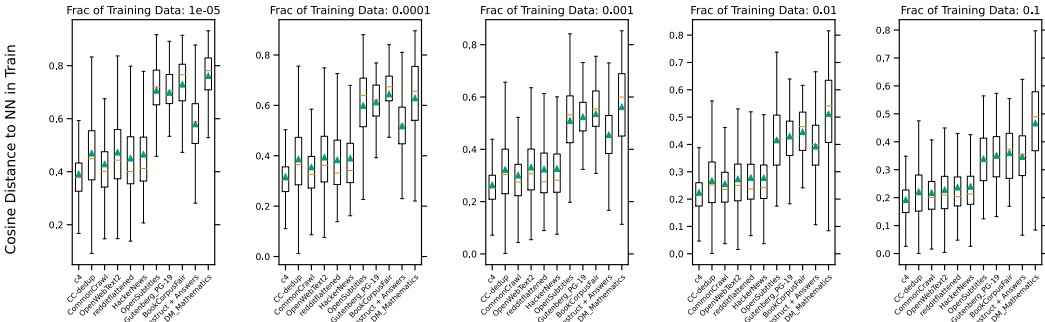

Figure A10: Investigating whether Figure 5 changes if we change training set size set size. In the figure above, each plot randomly samples a fraction of the training set (the fraction is denoted by the title of the plot). We see that the relative ranking of the validation sets generally remains the same, and there is consistently a jump between web-derived and web-independent validation sets.

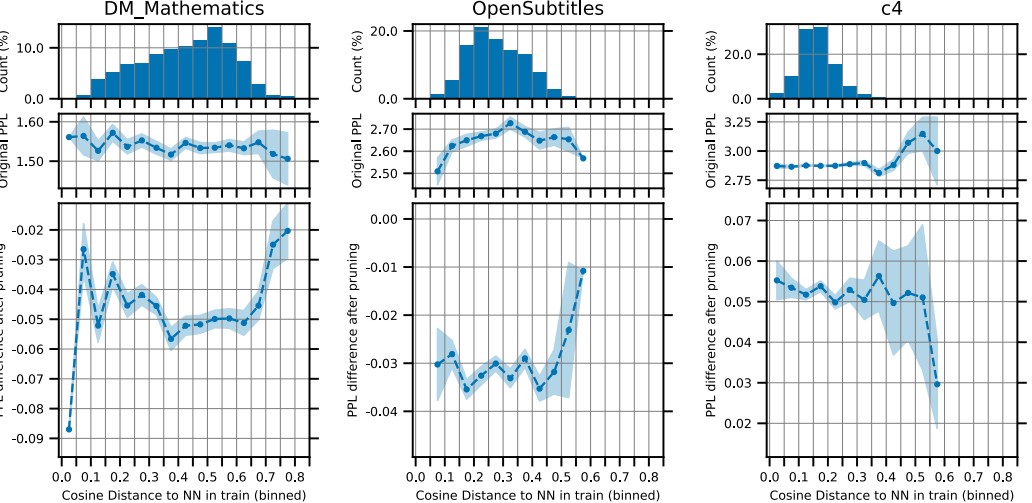

Figure A11: (Top): Histogram of cosine distance to nearest neighbor in train. Within each bin, we show the mean original perplexity (middle) and mean difference in perplexity after data selection (bottom), for DM_Mathematics (left), OpenSubtitles(middle), and C4 (right). We note that points in the C4 validation set closest to the training set are both "easy" (perhaps because of proximity to training points) and are affected the most by data selection. We do not see this trend for non-web snapshot validation sets such as DM_Mathematics and OpenSubtitles.

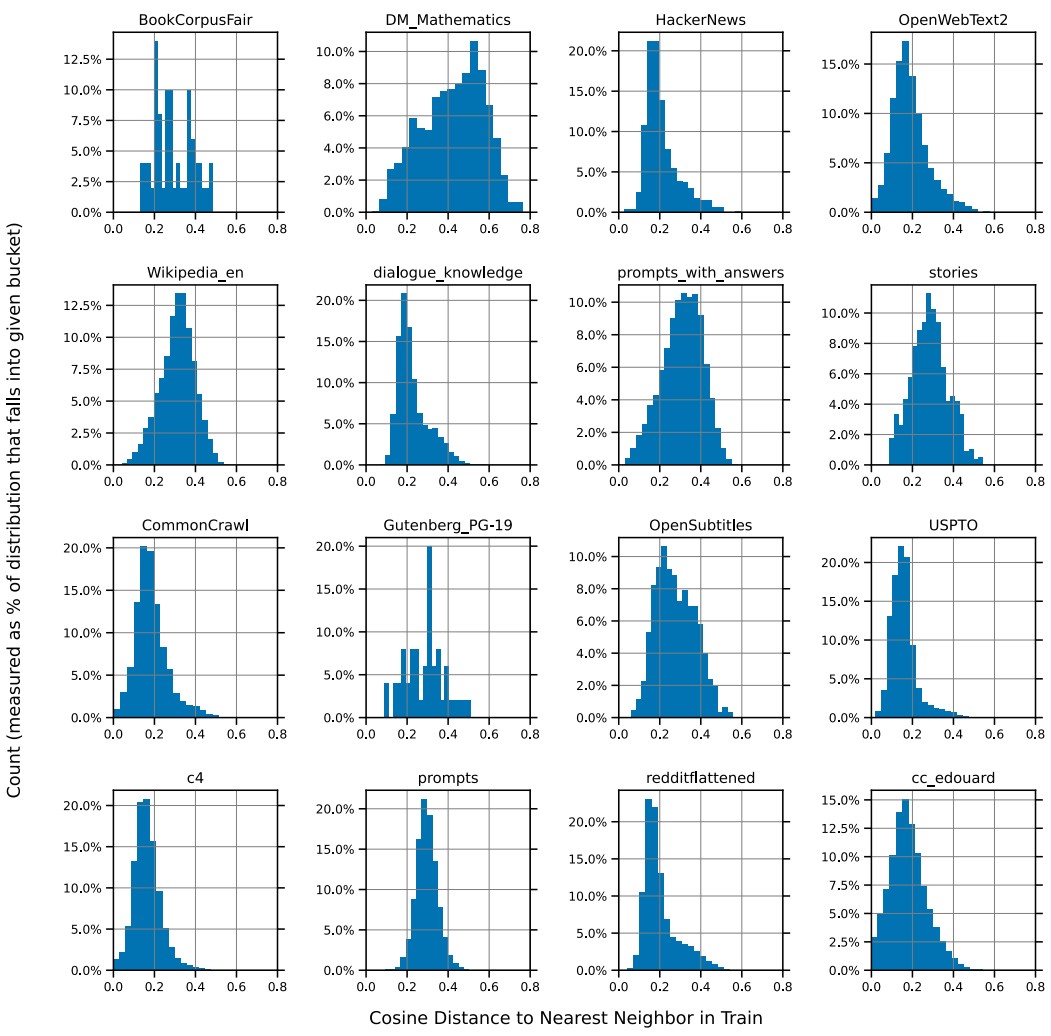

Figure A12: Distribution of cosine distance to nearest neighbor in the training set, for each individual validation set.

## A.6 Further investigation of repeating tokens

In this section, we investigate whether the findings from Section 4.2 hold across model scale, data selection ratio (e.g. number of epochs), and data selection method.

**Across data selection methods**: We first take the same configuration as Section 4.2, where we have a starting source dataset of 40B tokens, use each of our data selection methods with $R = 0.25$ to select a subset of documents, and repeat over these documents until we reach the target token budget of 40B tokens. Note that this is at the 1.3B model scale. In Figure A13 we see that repeating data selected by both SemDeDup and SSL prototypes also outperforms randomly selecting new data. However, we quickly notice that for *fixed* data selection strategy (e.g. *fixed* column in Figure A13), repeating tokens either outperforms or matched selecting new tokens. In other words: cleverly repeating tokens can outperform randomly selecting new tokens, but if we fix the data selection strategy (random, SemDeDup, SSL prototypes, or D4) then it is usually preferable to select new tokens. We also note in Figure A16 that D4 outperforms other methods, although by a smaller margin than in the fixed-compute regime.

**Across model scale and data selection ratio**: We fix our data selection strategy as D4 as done in Section 4.2, but attempt repeating tokens across 3 model scales (125M, 1.3B, and 6.7B), and across

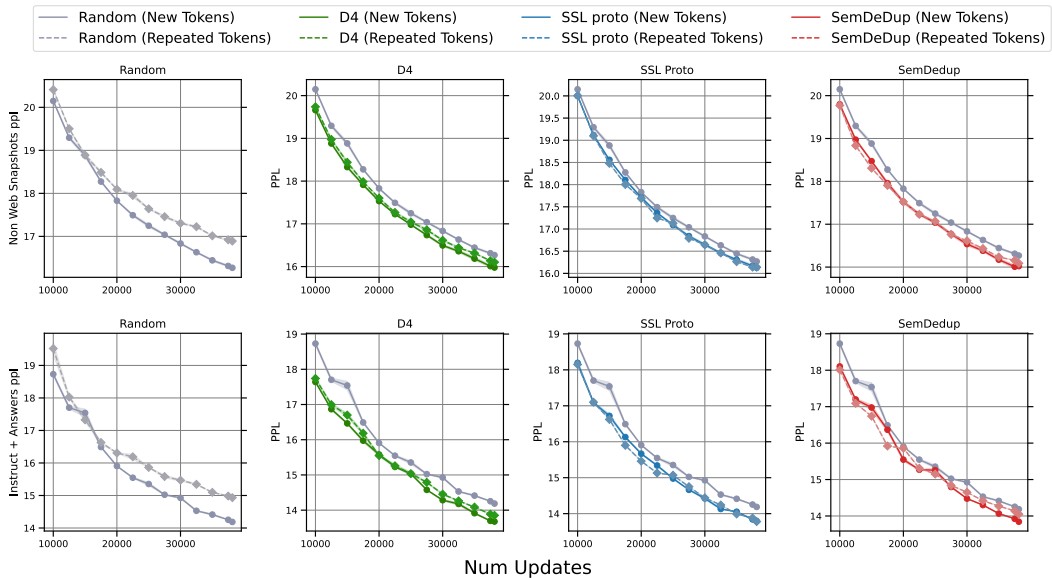

Figure A13: Effect of repeating tokens across data selection methods over training. X-axis denotes the number of updates, and the y-axis denotes average perplexity across non-web-snapshot validation sets (top row) and Instruct OPT (bottom row). Each column in the plot above denotes a different data selection method. Within each column: (1) the gray line denotes baseline training, (2) the colored-dashed line denotes repeating tokens via the specified data selection method, and (3) the colored-solid line denotes selecting new tokens via the specified data selection method. Repeating data is generally worse than selecting new data for a *fixed data selection method* (e.g., fixed column).

data selection ratios ($R = 0.5$ and $R = 0.25$). We see in Figure A15 that repeating data with D4 outperforms randomly selecting new tokens across all model scales and choice of $R$.

We note that for fixed $R$, different data selection methods will choose subsets of the source dataset that contain different amounts of tokens. This means that different data selection methods will epoch a different number of times. For example, for a 1.3B OPT model 40B token budget training run, if randomly repeating data with $R = 0.25$ chooses a subset with 10B tokens and D4 with $R = 0.25$ chooses a subset with 15B tokens, then the random run will epoch 4 times while the D4 run will epoch 2.67 times. To show this more clearly, we plot 1.3B and 6.7B repeated data runs with the x-axis changed to number of epochs in Figure A14. We see that up to roughly 2 epochs of data chosen with D4 significantly outperforms randomly selected new data; however, close to 5 epochs leads to worse performance.

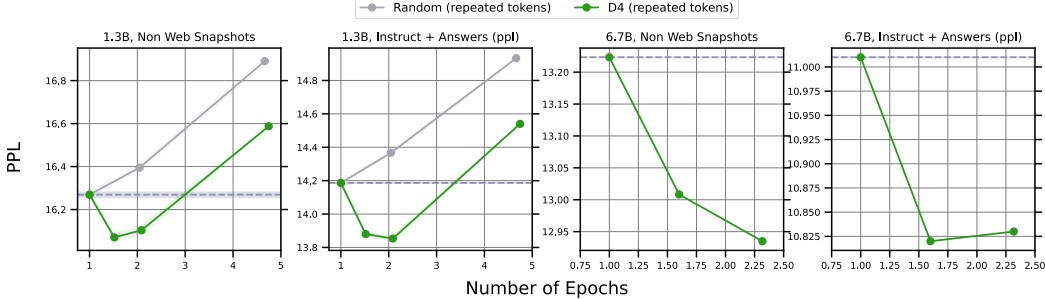

Figure A14: Comparison of repeating tokens with D4 (pink line), randomly selecting new tokens (horizontal dashed gray line), and randomly repeating data (gray line). We see with different epoch numbers. The y-axis denotes perplexity, and x-axis denotes number of epochs.

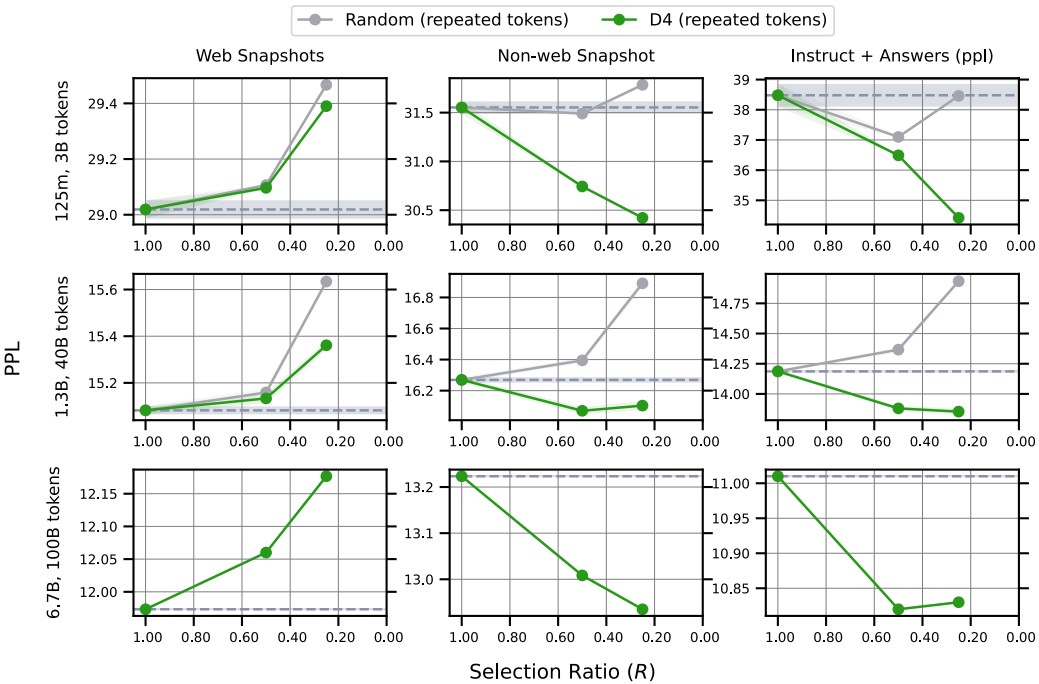

Figure A15: Comparison of repeating tokens with D4 (pink line), randomly selecting new tokens (horizontal dashed gray line), and randomly repeating data (gray line). We see across model scales (top: 125M trained on 3B tokens; middle: 1.3B trained on 40B tokens; bottom: 6.7B trained on 100B tokens) and data selection ratios, repeating data selected by D4 outperforms randomly selecting new data.

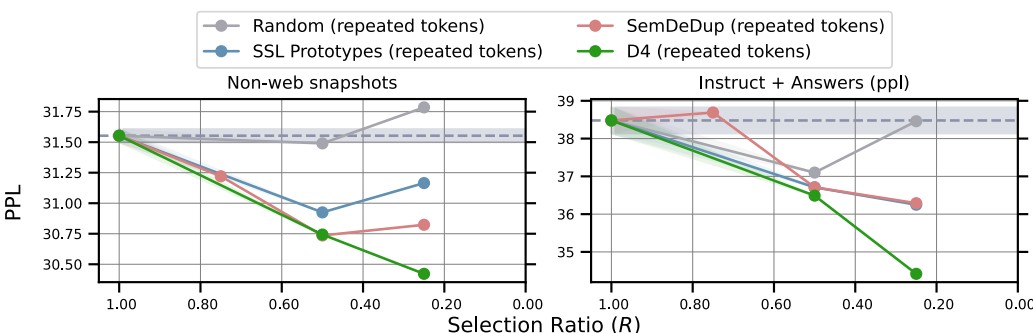

Figure A16: Comparison data selection methods when repeating data at the 125M, 3B token budget scale. The x-axis is data selection ratio $R$, and the y-axis is average perplexity on validation sets. We observe that selecting data to repeat via D4 outperforms other data selection methods, especially at low selection ratios $R$ (note that low selection ratios in the fixed-data regime correspond to more epochs).

## A.7 Choice of Embedding Space

All data selection methods we employ rely heavily on the quality of the underlying embedding space. We qualitatively analyzed the embedding produced by the last-token last-layer OPT 125M model and observed a bias towards end-of-document format. For example, if documents all end with an email or a standard phrase ("Buy our product today!"), then these documents would be clustered together. This likely helps detect templates (since templates tend to end their text in very similar ways), but has

clear pitfalls — for example, if we took thousands of wikipedia articles about unrelated topics and appended the same email at the end of each article, they might be clustered together.

Motivated by this, we briefly experiment with different embedding spaces and discuss our results in this section.

### A.7.1 SentenceTransformer models

BERT embeddings have generally been used to accomplish various NLP tasks, because BERT (unlike GPT/OPT) is able to attend to every token in the input when producing an embedding (BERT is a encoder-decoder model, while OPT/GPT are decoder only). While there are numerous BERT-style models available, we hoped to achieve an embedding space that focused on semantic similarity. Thus, we opted to use the widely popular SentenceTransformer models [3], which are BERT-style models finetund specifically >1B text similarity pairs. We choose the top model on the SentenceTransformer leaderboard (all-mpnet-base-v2) and the smallest well-performing model (all-Mini-LM-v6). Note that these models have max context length of 256 and 384 (respectively), and we stuck with the SentenceTransformer default of truncating inputs to fit the max sequence length (i.e. these embeddings only consider the beginning of documents).

We observe, in Figure A17 that at small model scales, sentence transformer embedding spaces outperforms the OPT embedding space. Given these initial results, we took our most overall-all efficient embedding space at the 1.3b model scale ("all-mini-lm-v6") and ran a 6.7b training run with it. Surprisingly, we observed that at larger model scale, the OPT embedding space outperforms the "all-mini-LM-v6" embedding space. Given that the difference between "all-mini-LM-v6" and "all-mp-net-base-v2" is generally small (see Figure A17), we also expect the OPT embedding space to beat "all-mpnet-base-v2" at the 6.7b, although we were not able to complete this run due to compute restrictions. We see the same trend when we consider overall and naive efficiency of using D4 with different embedding spaces in Figure A18.

In an effort to understand why SentenceTransformer embedding spaces perform worse at larger model scales, we qualitatively analyze the clusterings with each SentenceTransformer embedding space. We find that using D4 with "all-mp-net-base-v2" and "all-mini-lm-v6" disproportionately prunes long documents. We hypothesize that this is because sentence transformer models are trained and finetuned on actual sentence pairs, which very rarely saturate the max context length of the model. This might result in all "long" documents (or at least any input that is max-context-length size) seem out-of-distribution to the model. We guess that this results in long documents being clustered together, and therefore disproportionately affected during pruning. This might be especially relevant in domains like Wikipedia articles, where headers and introductions look semantically similar, but the actual content (past the first max-context-length tokens) is very different.

In an effort to circumvent this problem, we tried two approaches at a small model scale:

- M1: Chunking long documents into max-context-length chunks, and averaging all-mini-LM-v6 embeddings across chunks to produce a final document embedding.

- M2: Using Contriever [22] embeddings, where we chose the Contriever model because it is trained to determine if two sentences are from the same document, and therefore should be agnostic to position within a document.

Both in terms of perplexity improvement at the end of training (see Figure A19) and efficiency (see Figure A18) we do not observe a significant difference between the OPT embedding space and embedding spaces M1 and M2 at the small model scale (125 million parameters). We note that M1 and M2 are significantly worse than the all-mp-net-base-v2 and all-mini-LM-v6 at small scales **and** suffer from the same problem of pruning away long documents (compared to the OPT embedding space), so we expect these models to under-perform the OPT embedding space at the 6.7b scale.

---

[3] https://www.sbert.net/docs/pretrained_models.html

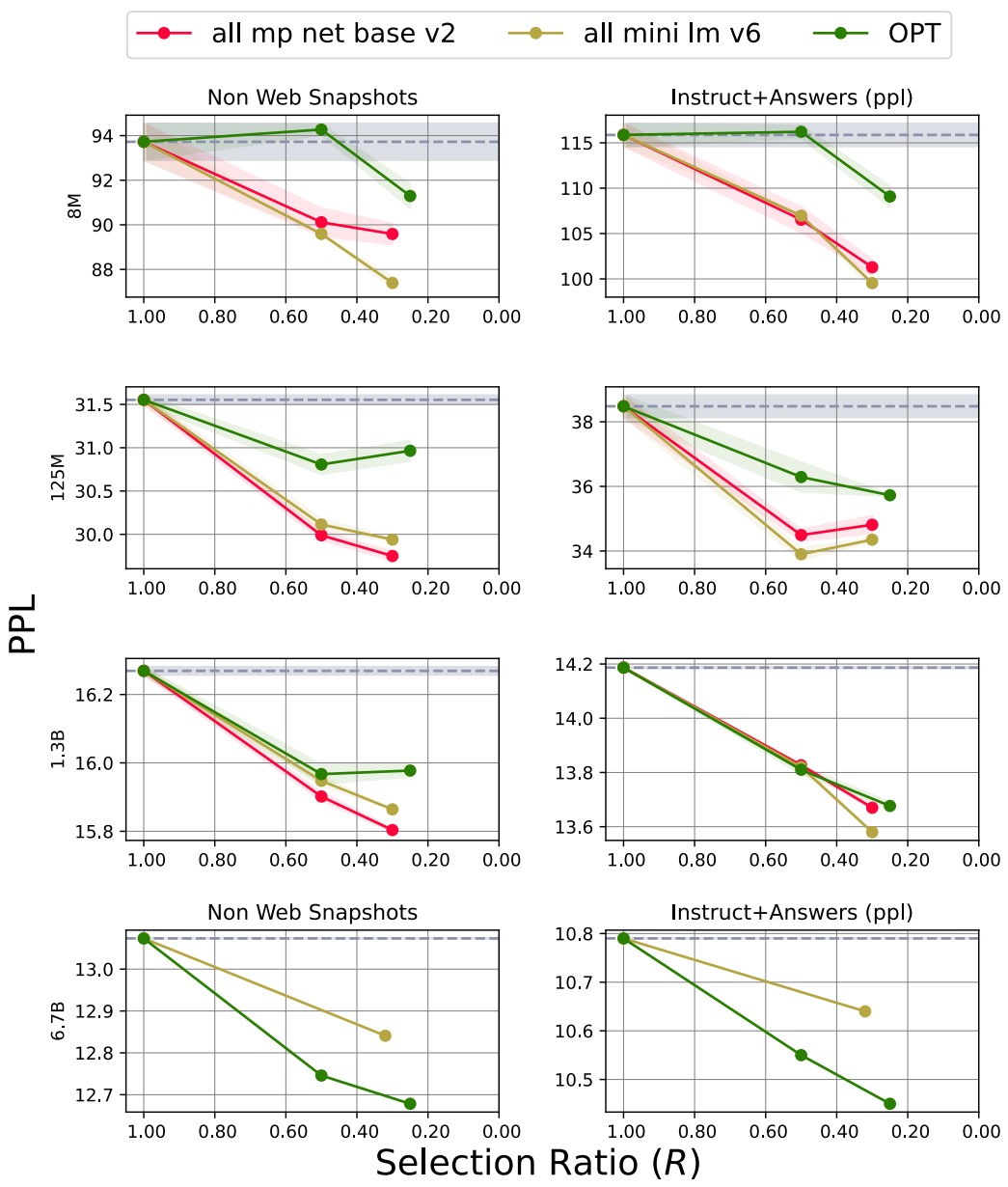

Figure A17: Perplexity (y-axis) versus selection ratio $R$ (x-axis) for different embedding spaces, when selecting data via D4. Across different 8m (top), 125m (middle) and 1.3b (bottom) model scales, we see that the SentenceTransformer embedding spaces outperform the OPT embedding space, but at the 6.7b model scale, we see that the OPT embedding space begins outperforming the all Mini LM v6 embedding space. We were unable to run an "all-mp-net-base-v2" 6.7b experiment due to compute restrictions, but we note that the difference between "all-mini-lm-v6" and "all-mp-net-base-v2" across model scales and selection ratios is generally small, so we expect the OPT embedding space to outperform the "all-mp-net-base-v2" at the 6.7b scale.

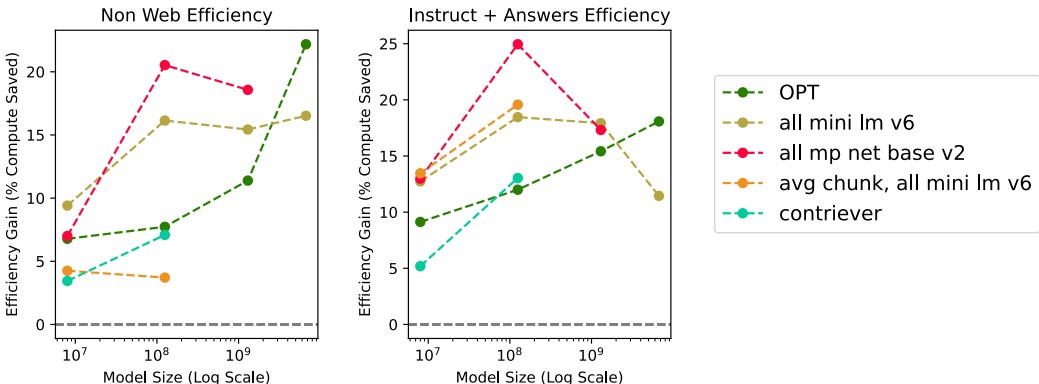

Figure A18: Comparison of naive efficiency for different embedding spaces, when using D4 as the data selection strategy. Similar to Figure A17, we see that all-mini-LM-v6 outperforms the OPT embedding space at small scale, but not at large (6.7b) model scale.

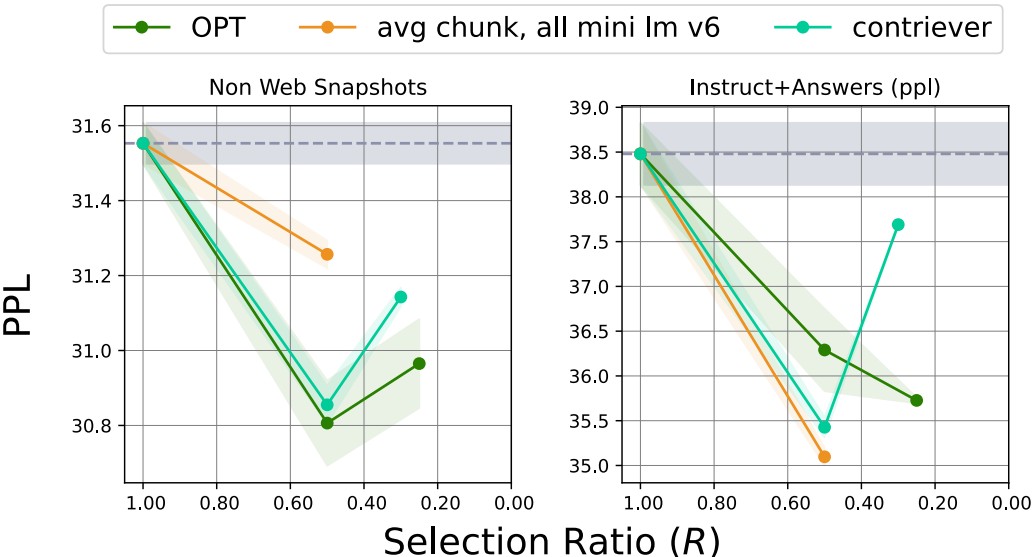

Figure A19: Comparison of embedding spaces M1 (averaging embedding of all-mini-LM-v6 across all chunks in a document, where a chunk is defined as 256 tokens) and M2 (embeddings from the Contriever model), with the OPT model embedding space, when using D4 as a the selection strategy. We note that neither embedding space signifigantly outperforms the OPT model embedding space at the 125M scale.

## A.8 Replicating Fixed Compute Results on C4

In this section, we briefly show our results for comparing data selecting methods at the 125M scale, where the pre-training dataset is the C4 [41] dataset instead of CC-dedup. We see in Figure A20 that D4 generally outperforms other methods. These initial experiments motivates us to try comparing data selection methods on more heavily filtered web-data (i.e. CC-dedup).

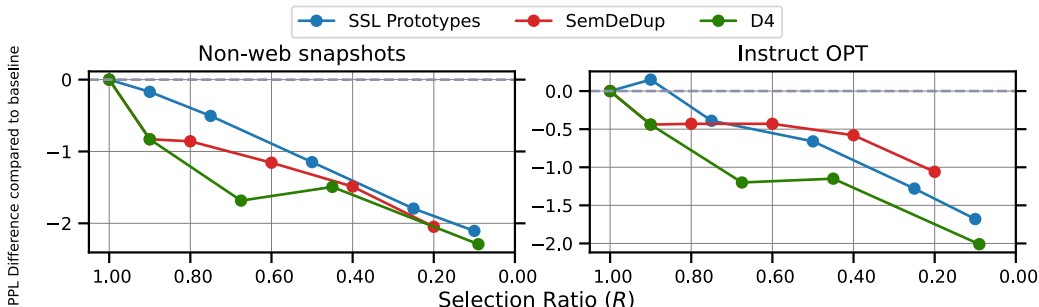

Figure A20: Comparison of data selection strategies with the OPT model embedding space, when using D4 as a the selection strategy, when using C4 as the starting training dataset. The x-axis is selectoin ratio $R$, and the y-axis is perplexity difference compared to baseline (the horizontal gray dotted line at 0.0 represents our baseline i.e. when no data selection is done), so **lower is better**. Notice that D4 and SemDeDup match at 90%, because we use $R_{dedup} = 0.9$ and varied $R_{proto}$ for this experiment.

## A.9 Investigating Duplicate-Driven Clusters

In this subsection, we present a few examples of duplicate-driven clusters, which are clusters that are very dense and near centroids. We find that these clusters tend to be filled with semantic duplicates and/or duplicated text. We generally can find such extreme duplicate-driven clusters by looking at clusters whose standard deviation of cosine distance to cluster centroid is less than 0.03. This is essentially looking at clusters in the lower tail of the empirical CDF in Figure 7 (brown line). We present a few examples of such clusters below:

Table A3: Nearest Neighbors to Cluster Centroid 682

| Cosine Distance to Centroid | Raw Text |
| --- | --- |
| 0.03581655 | The USGS (U.S. Geological Survey) publishes a set of the most commonly used topographic maps of the U.S. called US ......... may have differences in elevation and topography, the historic weather at the two separate locations may be different as well. |
| 0.03584063 | The USGS (U.S. Geological Survey) publishes a set of the most commonly used topographic maps of the U.S. called US ......... may have differences in elevation and topography, the historic weather at the two separate locations may be different as well. |
| 0.036803484 | The USGS (U.S. Geological Survey) publishes a set of the most commonly used topographic maps of the U.S. called US ......... may have differences in elevation and topography, the historic weather at the two separate locations may be different as well. |
| 0.037270606 | Search Near Clinton County, OH: Trails National and State Parks City Parks Lakes Lookouts Marinas Historical Sites The USGS (U.S. Geological ......... may have differences in elevation and topography, the historic weather at the two separate locations may be different as well. |

Table A4: Nearest Neighbors to Cluster Centroid 975

| Cosine Distance to Centroid | Raw Text |
| --- | --- |
| 0.011662006 | The American Way, Inc. The American Way, Inc. is a suspended Californian business entity incorporated 19th August 1949. is listed as ......... for bulk data downloadsI want to request the removal of a page on your websiteI want to contact California Explore |
| 0.012483656 | John St-Amour, Inc. John St-Amour, Inc. is a suspended Californian business entity incorporated 5th October 1962. is listed as the agent ......... for bulk data downloadsI want to request the removal of a page on your websiteI want to contact California Explore |
| 0.012564898 | Joseph E. Barbour, Inc. Joseph E. Barbour, Inc. is a suspended Californian business entity incorporated 27th January 1959. is listed as ......... for bulk data downloadsI want to request the removal of a page on your websiteI want to contact California Explore |
| 0.012756169 | The Jolly Boys, Inc. The Jolly Boys, Inc. is a suspended Californian business entity incorporated 4th March 1955. is listed as ......... for bulk data downloadsI want to request the removal of a page on your websiteI want to contact California Explore |

Table A5: Nearest Neighbors to Cluster Centroid 10715

| Cosine Distance to Centroid | Raw Text |
| --- | --- |
| 0.035506427 | Search hundreds of travel sites at once for hotel deals at Hotel Olympic Kornarou Square 44, Heraklion, Greece 34 m Bembo Fountain 262 ......... hundreds of travel sites to help you find and book the hotel deal at Hotel Olympic that suits you best. |
| 0.036230028 | Search hundreds of travel sites at once for hotel deals at Hotel Estrella del Norte Juan Hormaechea, s/n, 39195 Isla, Cantabria, ......... travel sites to help you find and book the hotel deal at Hotel Estrella del Norte that suits you best. |
| 0.036280274 | Search hundreds of travel sites at once for hotel deals at H10 Costa Adeje Palace Provided by H10 Costa Adeje Palace Provided ......... travel sites to help you find and book the hotel deal at H10 Costa Adeje Palace that suits you best. |
| 0.036827266 | Search hundreds of travel sites at once for hotel deals at Hotel Miguel Angel by BlueBay Calle Miguel Angel 29-31, 28010 ......... sites to help you find and book the hotel deal at Hotel Miguel Angel by BlueBay that suits you best. |

Table A6: Random Examples from Cluster 695

| Cosine Distance to Cluster Centroid | Raw Text |
| --- | --- |
| 0.044178426 | Eastern Florida State College nutritional sciences Learn about Eastern Florida State College nutritional sciences, and registering for electives. Which college degrees ......... System (IPEDS). If any stats on Hagerstown Community College career planning are incorrect, please contact us with the right data. |
| 0.056984067 | Albany State University introduction to business Find info concerning Albany State University introduction to business, and registering for elective discussion sections ......... If any stats on Warren County Community College plant science major are incorrect, please contact us with the right data. |
| 0.0534693 | Baldwin Wallace University cost per unit Learn about Baldwin Wallace University cost per unit, submitting required application forms, and follow-up scheduling. ......... (IPEDS). If any stats on San Jose State nursing degree programs are incorrect, please contact us with the right data. |
| 0.06892538 | Niagara University managerial accounting Information about Niagara University managerial accounting, and registering for elective lectures. Which college degrees give you the ......... System (IPEDS). If any stats on Midwestern University pharmacy tech program are incorrect, please contact us with the right data. |
| 0.07246786 | Fanshawe College app download Learn about Fanshawe College app download, and registering for elective discussion sections and seminars. Which college degrees ......... Data System (IPEDS). If any stats on Stratford University cell biology are incorrect, please contact us with the right data. |
| 0.07147932 | Standish Maine Licensed Vocational Nurse LVN Jobs Find out about Standish, ME licensed vocational nurse LVN jobs options. It's a smart ......... (IPEDS). If any stats on William Jewell College medical insurance coding are incorrect, please contact us with the right data. |

Table A7: Random Examples from Cluster 8342

| Cosine Distance to Cluster Centroid | Raw Text |
| --- | --- |
| 0.027729392 | Seenti - Bundi Seenti Population - Bundi, Rajasthan Seenti is a medium size village located in Bundi Tehsil of Bundi district, Rajasthan ......... 6 months. Of 186 workers engaged in Main Work, 63 were cultivators (owner or co-owner) while 0 were Agricultural labourer. |
| 0.036407113 | Kodunaickenpatty pudur - Salem Kodunaickenpatty pudur Population - Salem, Tamil Nadu Kodunaickenpatty pudur is a large village located in Omalur Taluka of ......... 6 months. Of 3523 workers engaged in Main Work, 1500 were cultivators (owner or co-owner) while 1533 were Agricultural labourer. |
| 0.017463684 | Chhotepur - Gurdaspur Chhotepur Population - Gurdaspur, Punjab Chhotepur is a medium size village located in Gurdaspur Tehsil of Gurdaspur district, Punjab ......... 6 months. Of 677 workers engaged in Main Work, 123 were cultivators (owner or co-owner) while 142 were Agricultural labourer. |
| 0.02616191 | Maksudanpur - Azamgarh Maksudanpur Population - Azamgarh, Uttar Pradesh Maksudanpur is a small village located in Sagri Tehsil of Azamgarh district, Uttar ......... 6 months. Of 22 workers engaged in Main Work, 14 were cultivators (owner or co-owner) while 0 were Agricultural labourer. |
| 0.028420448 | Karambavane - Ratnagiri Karambavane Population - Ratnagiri, Maharashtra Karambavane is a medium size village located in Chiplun Taluka of Ratnagiri district, Maharashtra ......... 6 months. Of 444 workers engaged in Main Work, 116 were cultivators (owner or co-owner) while 214 were Agricultural labourer. |
| 0.037917078 | Barda - Purba Medinipur Barda Population - Purba Medinipur, West Bengal Barda is a large village located in Egra - I Block ......... 6 months. Of 1182 workers engaged in Main Work, 278 were cultivators (owner or co-owner) while 252 were Agricultural labourer. |

Table A8: Nearest Neighbors to random validation point in C4

| Cosine Distance | Raw Text |
|---|---|
| 0.0(original validation text) | Offers two child care opportunities to Charles County citizens— the Port Tobacco Onsite Child Care Program and the Before and After School Child Care Program (BASCC). Supports parents through home visits to first time parents and by helping them search for child care, find resources for a child with social, emotional . . . . . . . . Special needs kids. Free to look, a fee to contact the providers. Hotline is staffed by highly-trained and friendly Child Care Consumer Education Specialists who offer both parents and providers invaluable information about child care, and referrals to local Child Care Resource and Referral agencies where they can receive individualized assistance. |
| 0.12867724895477295 | Child Care Options is a program of Options Community Services , a non-profit registered charity dedicated to making a difference in the South Fraser Region. Options is committed to empowering individuals, supporting families and promoting community health. Funding for Child Care Options is provided through British Columbia's Ministry of Children . . . . . . . Rock. Child Care Options links families and child care providers in the communities of Delta, Surrey and White Rock by offering free consultation, support and child care referral services and subsidy support to parents seeking child care. Child care providers are supported through information, outreach, resource library, networking, and learning opportunities. |
| 0.15080827474594116 | Below are links to child development resources, both from within the department and from external sources. Child Development Division Publications Publications that can help you will help you follow your child's development (from birth to age five) so you can identify and address any issues early on. Resources to help you understand children's . . . . . . . . families to local resources and services. Specialists are available from 9 AM to 6 PM Monday – Friday. Services are confidential. Caregivers can also visit http://www.helpmegrowvt.org/families.html to learn more about child development, discover developmental tips, and watch videos demonstrating children's developmental milestones (click a button to choose your child's age). |
| 0.15738284587860107 | National Domestic Violence Hotlines Programs that provide immediate assistance for women and men who have experienced domestic abuse which may include steps to ensure the person's safety; short-term emotional support; assistance with shelter; legal information and advocacy; referrals for medical treatment; ongoing counseling and/or group support; and other related services. Hotline . . . . . . . . RP-1500.1400-200) www.thehotline.org/ Toll Free Phone: 800-799-SAFE URL: https://www.thehotline.org/ Eligibility: Anyone affected by relationship abuse. Services Provided: Available 24/7/365 via phone, TTY, and chat. Provides lifesaving tools and immediate support to enable victims to find safety and live lives free of abuse. Highly trained, experienced advocates offer support, crisis intervention, education, safety planning, and referral services. |

Table A9: Nearest Neighbors to random validation point in USPTO

| Cosine Distance | Raw Text |
|---|---|
| 0.0(original validation text) | SONET (Synchronous Optical NETwork) is a North American transmission standard for optical communication systems. SDH (Synchronous Digital Hierarchy), a European transmission standard, is a minor variant of SONET. SONET defines a hierarchy of electrical signals referred to as Synchronous Transport Signals (STS). The STS hierarchy is built upon a basic signal . . . . . . . the corresponding row and column numbers may include up to 18 comparison operations, which are onerous to implement, for example, in terms of the required logic circuitry. This problem is exacerbated at the upper levels of the STS hierarchy, where processing of multiple pointer values per data frame is performed. |
| 0.1998944878578186 | US20080109728A1 - Methods and Systems for Effecting Video Transitions Represented By Bitmaps - Google Patents Methods and Systems for Effecting Video Transitions Represented By Bitmaps Download PDF David Maymudes Multi-media project editing methods and systems are described. In one embodiment, a project editing system comprises a multi-media editing application that is configured to . . . . . . . . synchronization models for multimedia data US20120206653A1 (en) 2012-08-16 Efficient Media Processing US6658477B1 (en) 2003-12-02 Improving the control of streaming data through multiple processing modules US6212574B1 (en) 2001-04-03 User mode proxy of kernel mode operations in a computer operating system US7752548B2 (en) 2010-07-06 Features such as titles, transitions, and/or effects which vary according to positions |
| 0.21122217178344727 | Both the Ethernet II and IEEE 802.3 standards define the minimum frame size as 64 bytes and the maximum as 1518 bytes. This includes all bytes from the Destination MAC Address field through the Frame Check Sequence (FCS) field. The Preamble and Start Frame Delimiter fields are not included when . . . . . . . frame. Dropped frames are likely to be the result of collisions or other unwanted signals and are therefore considered invalid. At the data link layer the frame structure is nearly identical. At the physical layer different versions of Ethernet vary in their method for detecting and placing data on the media. |
| 0.2133803367614746 | A byte is a group of bits, usually eight. As memory capacities increase, the capacity of chip cards is often quoted in bytes rather than in bits as in the past. |