# OpenReview forum: "D4: Improving LLM Pretraining via Document De-Duplication and Diversification"
_NeurIPS.cc/2023/Track/Datasets_and_Benchmarks — NeurIPS 2023 Datasets and Benchmarks Poster_

### Official Review · Reviewer_XL6a · 2023-06-22
**Simple idea for effective data selection, extensive experiments, limited novelty.**

**Rating:** 6
**Confidence:** 4

**Strengths:**

- A simple method that gets better value out of text data for pre-training LLMs.
- Extensive experiments
- Mostly well-written


**Additional Feedback:**

-

**Clarity:**

- The main thing I find unclear about this work is the definition of a document. Pre-trained LMs are typically trained on running texts split to sequences of N tokens. Do the authors consider each such sequence as a document? Or perhaps longer spans? This is partly addressed in the limitations section, but I am still not sure what the authors actually did.
- Instruction Tuning Perplexity: I am not sure I follow on which text you are computing perplexity. Is this the set of instructions themselves?
- What is the overhead ratio in line 169 & 171?
- What is the difference between the naive and overall efficiency gains in Fig 4?
- Related to the first point, what does the histogram in Fig5 (top right) represent? The proportion of documents with each similarity score?

Typos and such:
- Fig1: (a) I believe you mixed the words “right” and “left” in the caption (b) the figure appears on page 3 but is first mentioned only on page 5
- #252: ”we see that demonstrate that“


**Correctness:**

The claims are mostly correct, but I have one concern. The split between web and non-web data seems rather coarse-grained to me. For instance, what makes hackerNews web-derived while open-subtitles web-independent (what does it even mean to be web independent?). I agree that generally data collected from the web might be noisier than data that is crafted by experts, but the claim is too strong (if only because web data also contains some high quality data).


**Documentation:**

N/A

**Limitations:**

The authors address most of the main limitations properly.


**Opportunities For Improvement:**

- The proposed method is a bit incremental–running two existing methods in a pipeline. In general there is limited novelty in this work, mostly a large amount of (very expensive) experiments. The main interesting insight is the finding that running two epochs on carefully selected data can outperform one epoch over randomly selected one, but this is just a single experiment (although it is consistent across the training process, Fig3), which might not generalize.
- I don’t fully agree with the web/non-web part (see below).


**Relation To Prior Work:**

This work builds on two recent efforts – SemDeDup and Prototypicality, and touches on many other related methods.


**Summary And Contributions:**

This paper presents a data filtering method for pre-training language models. The method  works by running two existing approaches (SemDeDup and Prototypicality) in a pipeline. The authors train several LMs using the proposed strategy, showing they can save up to 20% of the training time compared to randomly selecting data. They also show that the selected data can be duplicated (i.e., run 2 epochs on it) and still show gains.

---

> ### Author Response · Authors · 2023-08-21
>
> Dear Reviewer XL6a,
>
> Thank you for your time to review our paper and for your numerous comments and suggestions. We are very glad that you found our experimental setup to be comprehensive, and our work to be well-written. We have adjusted several sections in our revised manuscript following your observations regarding the clarity and correctness. In particular, here are the comments we addressed:
>
> > The split between web and non-web data seems rather coarse-grained to me
>
> We thank the reviewer for pointing this out. The purpose of this split was to provide an easy conceptual framework for readers to understand why we empirically see that validation sets from web crawls behave differently than other validation sets. However, we agree with the reviewer that the split is rather coarse-grained. In reality, there is a smooth transition from web snapshot data => web-derived data => web-independent data. Web-snapshot data include things like CommonCrawl, C4, CC-dedup which are representative of what you would find in a CommonCrawl snapshot. Web-derived data are focused portions of the web, for example only reddit, or only Wikipedia, or only HackerNews. Web-independent data is data that is generated independently of the web like instructions + answers or patent applications (i.e. if you randomly sample CommonCrawl snapshots, you are very unlikely to find instances of web-independent data). We have changed Figure 5 and the text in Section 4.4.1 to reflect this smooth transition more accurately.
>
> To be clear, when we report results:
>
> Web snapshots = C4, CC-dedup (same distribution as our training set), CommonCrawl
>
> Non-web snapshots = web-derived + web-independent data
>
> This is also emphasized in appendix Section A.1.4.
>
> > The main thing I find unclear about this work is the definition of a document.
>
> We have modified Section 3 of the paper  in order to make this more clear to the reader. The summary of these changes is the following:  A document corresponds to the crawled content of a single web URL. We consider documents as the atomic units of the selection method - each document is either selected or not selected. We do not concern ourselves with how the trainer converts these sequences into batches of token sequences - we treat it as a black box that takes a stream of documents defined by our data selection method and returns a trained model after a given number of model updates.
>
>
> > Instruction Tuning Perplexity: I am not sure I follow on which text you are computing perplexity.
>
> We thank the reviewer for pointing this confusion out. We are simply measuring the perplexity of another validation set, which is filled with prompts + answers from a standard instruction tuning dataset (the instruction tuning dataset used for OPT IML [1]). We have clarified the description of the Instruction Tuning Perplexity set in Section 3.3, and provided examples from the validation set in appendix Table A2 (and we have modified appendix section A.1.4 to give more details about this validation set).
>
> > What is the overhead ratio in line 169 & 171?
>
> Thank you for finding this inconsistency in naming our parameters. In our revised manuscript we replaced “overhead ratio” with “selection ratio”, which is exactly the % of documents chosen from the source dataset. We have emphasized this definition at the beginning of Section 3.
>
> > What is the difference between the naive and overall efficiency gains?
>
> Thank you for pointing out that we have missed introducing these terms properly. We define overall efficiency gain simply as naive efficiency gain minus the cost of computing the metric. We have corrected this in our revised manuscript and hope that reading this version will clarify this question.
>
>
> We thank the reviewer again for their suggestions and we hope that our changes have helped to answer their questions. If the reviewer found that we addressed their comments appropriately, we kindly ask them to consider increasing the review score accordingly.
>
>
> Citations:
>
> [1]: Iyer, Srinivasan, et al. "Opt-iml: Scaling language model instruction meta learning through the lens of generalization." arXiv preprint arXiv:2212.12017 (2022).

---

> > ### Comment · Reviewer_XL6a · 2023-08-23
> >
> > Thank you for the clarification and the updates to the paper.
> >
> > I will keep my initial score.

---

### Official Review · Reviewer_nVTQ · 2023-07-10
**Interesting analyses of deduplication in pretraining**

**Rating:** 5
**Confidence:** 4
**Clarity:** Paper is well written and easy to fol…

**Strengths:**

1. Authors aim to improve quality of language models by deduplicating training samples. It's topic of great importance, findings may be widely useful.
1. Authors propose new deduplication method called D4, which according to their analyses is better than ones presented before.
1. There are many experiments showing usefulness of using D4 when ones goal is to achieve low perplexity on target task (both in limited compute and limited data regimes).
1. Authors show their method computational cost is low enough to be useful in practice.
1. There is presented interesting analyse about difference between web and non-web corpora. Unfortunately, it's methodically flawed (see Opportunities For Improvement), nevertheless it's an interesting topic.


**Additional Feedback:**

I personally find this work really interesting.

**Correctness:**

I don't have any concerns except my concerns to the methodology used in section 4.4.1 (point 3. in Opportunities For Improvement). Main idea and crucial experiments seems correct.

**Documentation:**

No data, code or anything else requiring documentation was released.

**Ethics:**

No concerns.

**Limitations:**

Many limitations were pointed out in the paper and I agree with them all, however I'd put more emphasis on limitations in context of used training dataset. Proposed method works well in context of a particular dataset created with particular pipeline. It would be interesting to see how the method applies to some other corpus, for instance C4.

The work is also limited in more general scope: aim when proposing D4 method is to produce corps, which maximizes score on set of NLP tasks, not to create corpus which realistically represents language (for instance removing documents from dense areas in embedding space may result in some skew in a model).

**Opportunities For Improvement:**

1. The biggest problem in this work is usage of Instruct OPT metric as the most important in all the considerations. In practice NLP community is more interested in scores on downstream tasks. Authors rarely present effects in terms of this metric. I understand the downstream scores have big variance and correlation between Instruct OPT and downstream accuracy was calculated, but still using Instruct OPT as the main metric greatly diminishes scientific importance of this work.
1. Section 4.3. Is embedding the first or the second step? In the first paragraph of this section the authors write ": first, embedding documents", however in second paragraph there is "For the second step, embedding 400B tokens". I'm quite confused.
1. Subsection 4.4.1, figure 5, left (and corresponding text). Average distance from validation embedding to its nearest training neighbour greatly depends of a dataset size (more documents -> more embeddings -> on average closest neighbour is closer). Hence the difference the authors show between internet and not-internet dataset may be caused by dataset size.
1. Colour selection in Figure 1. It's impossible to see where pink confidence interval ends if pink confidence interval overlaps with the red one.
1. Instruction Tuning Perplexity: I don't see information if it's in 0-shot or fine-tuned approach.
1. `redditflattened` term was introduced but never used.

**Relation To Prior Work:**

No concerns.

**Summary And Contributions:**

Authors provide new method for deduplicating corpus for language model pretraining and show its usefulness. They show it helps achieve lower perplexity on downstream task within the same training budget, lower computation cost to achieve the same quality model and helps to train high quality model in low-data regime.

---

> ### Author Response · Authors · 2023-08-21
> **Response Part 1**
>
> Dear Reviewer nVTQ,
>
> Thank you for your review, insightful comments, and the various suggested opportunities for improvement. We are glad that you found our work to tackle important and interesting questions. We have incorporated many of your points in our revised manuscript. Please find detailed responses to your suggestions below.
>
> > The biggest problem in this work is usage of Instruct OPT metric as the most important in all the considerations.
>
> We thank the reviewer for pointing this out, but we would like to clarify a possible confusion. We do not consider the Instruct OPT metric (instruction-tuning corpus perplexity) as our most important metric. We use a combination of 3 metrics that all lead to similar conclusions regarding D4: perplexity on the original OPT validation sets, perplexity on the OPT-IML instruction-tuning corpus, and downstream task accuracy. We have included additional details on the Instruct OPT dataset and its motivation in Section 3.3 of the revised manuscript, to emphasize this point.
>
> Additionally, we would like to point out that our results on all 3 of these metrics are statistically significant:
> For all Figure 2 (< 1.3B scale) results: we run pre-training with 3 random seeds and provide error bars
> For downstream accuracy results: To provide some context for our results on downstream task accuracy, we refer the reviewer to Figure 3. of the OPT paper (Zhang et al. 2022). This figure shows that 2-3% improvement in downstream task accuracy is about as much as 1) the maximum difference one can observe between the performance OPT and GPT3 at any given scale 2) how much GPT3 or OPT improve between 2.7B and 6.7B model scales.
>
> We use perplexity on an instruction-tuning corpus as an additional metric because it is the one of the main signals used to fine-tune language models for downstream-use cases (chatbots, assistants, etc.). Examples in this dataset include translation tasks (“translate this sentence from English to French”), general QA (“How does an electric car run on electricity?”), and programming tasks (e.g. “turn this natural language prompt into an SQL query”), among others. We have added examples of the instruction-tuning corpus in Table A2 in the appendix.
>
> Why should we use an instruction-tuning corpus in the first place? These types of natural instruction prompts mirror how people prompt language models in normal, everyday use-cases, thus aligning with what we use language models for in practice. Also, recent research (Gonen et al. 2022) shows that prompts on which a model has lower perplexity tend to result in higher task accuracy — indicating that the ability to model the prompt texts themselves is an important factor for good downstream performance. We also find that perplexity on instruction tuning data has a positive correlation with downstream accuracy (Figure 6).
>
>
> > Section 4.3. Is embedding the first or the second step?
>
> Section 3 starts by stating that all of our methods rely on an embedding step (L148-150), this is what we refer to as the “first step” in section 4.3. The reviewer rightly noted that starting the numbering at 1 in Section 3.4 can cause confusion, we have addressed this in our revised manuscript.

---

> > ### Author Response · Authors · 2023-08-21
> > **Response Part 2**
> >
> > > Average distance from validation embedding to its nearest training neighbor greatly depends of a dataset size
> >
> > We thank the reviewer for pointing this out. To clarify, in Figure 5, we keep the size of the training set *fixed*, while the sizes of different validation sets may vary.
> >
> > We were unsure whether “dataset size” in the reviewer’s comment refers to the validation set sizes or training set sizes, so we ran experiments two sets of experiments:
> >
> > If the reviewer was indicating that the difference between internet and non-internet validation sets in Figure 5 could be merely due to difference in size of validation sets: in Figure A8, we run an experiment where we keep all validation sets the same size (50 points, because the smallest validation set we use — BookCorpusFair — has size 50). If the original validation set is too big, we randomly sample 50 points from the validation set. We observe that the same distinction between internet and non-internet validation sets exist, even after fixing validation set sizes. We also observe in Figure A7 that arbitrarily changing the validation set sizes does not affect the mean distance to training nearest neighbor. Together, this indicates that Figure 5 is not due to differences in validation set size, but rather because some validation sets are truly semantically “closer” to the training distribution than others.
> >
> > If the reviewer was indicating that the difference in Figure 5 between internet and non-internet validations sets could be due to the size of the training set: we run an experiment in Figure A10  where we reproduce Figure 5 but reduce the training set size (we randomly sample a fraction of the training set), and then compute nearest neighbors in the training set for each validation set. We observe that across different training set sizes, the big difference between internet and non-internet validation sets remains consistent. As a sanity-check, we see in Figure A9 that reducing the size of the training set does generally increases the cosine distance of “nearest neighbor to train”, but this increase happens uniformly across validation sets. In other words, for any fixed training set size, the ordering of validation sets in Figure 5 generally remains the same.
> >
> > > Colour selection in Figure 1.
> >
> > We thank the reviewer for pointing out the confusing color selection, we have changed colors in Figure 1 (as well as the rest of the paper) to make the distinction between D4 and other methods more visually clear. Please let us know if the colors in the plots are still confusing or unreadable.
> > Note: we have swapped Figure 1 and Figure 2 so that the text is closer to the actual figures, so these changes are reflected in Figure 2.
> >
> > > Instruction Tuning Perplexity: I don't see information if it's in 0-shot or fine-tuned approach.
> >
> > We believe our definition of “Instruction Tuning Perplexity” was a bit unclear. We are simply measuring the perplexity of another validation set, which is filled with prompts + answers from a standard instruction tuning dataset (the instruction tuning dataset used for Instruct OPT). We are not doing instruction-tuning / fine-tuning, or doing k-shot evaluation. We have updated the definition in Section 3 to more explicitly state this and added example texts from this corpus in Appendix A.1.4 (Table A2)
> >
> > > redditflattened term was introduced but never used.
> >
> > We thank the reviewer for pointing this out. We did introduce the term `redditflattened` in Section 3.3 under the section “Validation Set Perplexity”, but we agree it was a bit buried with the text. We have modified the format so that it is easier to find this term.
> >
> >
> > >  It would be interesting to see how the method applies to some other corpus, for instance C4.
> >
> > We thank the reviewer for suggesting this, and have run additional experiments on C4. We have added these results in appendix section A.9. We observe in Figure A20 that the results are the same as on the CC-dedup dataset (that is, selecting via D4 significantly improves performance, and D4 outperforms SemDeDup and SSL prototypes).
> >
> > We chose the LLaMA pipeline in particular because it contains a MinHash-based deduping step, similarly to all other currently prevalent web-scale data pipelines. This in contrast to C4, which has a more lenient deduping mechanism (requiring exact matches of 3 sentences, see Raffel et al. 2020) that we expect to catch even less redundancy in the data. While we in fact started our experiments on C4 (we have added these experiments in appendix section A.9), we quickly moved on to the LLaMA pipeline as we believe 1) it is more challenging to improve on due to MinHash deduplication 2) it represents the data pipelines used in recent LLMs well.
> >
> > We would like to thank the reviewer for suggesting this experiment. These results help present a clearer empirical picture of the effect of D4 and bolster our main findings.

---

> > > ### Author Response · Authors · 2023-08-21
> > > **Response Part 3**
> > >
> > > >  aim when proposing D4 method is to produce corps, which maximizes score on set of NLP tasks, not to create corpus which realistically represents language
> > >
> > > We find the reviewer’s thoughts on motivation very much in-line with the Authors’ motivation for this work. In fact, this work was motivated by carefully observing our data and identifying “artifacts” of web-crawl data that contain a lot of redundancy and do not realistically represent language well. A few examples of this include templated text and advertisements (shoe ads, mattress ads, etc.). We agree that this has not gotten enough emphasis in the paper and addressed this issue by including a paragraph in Section 3.4. We also present some of these motivating examples in Appendix Tables A3, A4, A5, A6, and A7).
> > >
> > > We thank the reviewer again for their time. We hope that additional analysis and results we provided resolved some of the reviewer’s concerns, as well as that our responses and the revised manuscript helped to clarify some of the reviewer’s questions. If this was the case, we ask the reviewer to kindly consider increasing the review score accordingly.

---

> > > > ### Comment · Reviewer_nVTQ · 2023-08-30
> > > >
> > > > Thank you for the answers and changes introduced in the manuscript. I will keep my score.

---

### Official Review · Reviewer_qR3n · 2023-07-21
**A new data selection strategy by sequentially applying two existing data deduplication methods**

**Rating:** 6
**Confidence:** 2

**Strengths:**

The paper studies the important topic of improving data efficiency in large language model training via deduplication. As improving large language model pretraining gets increasingly expensive, data efficiency is essential for reducing the overall cost of training these models.

The experiments done on this paper was  repeated using several different language model architectures of varying sizes, and the experiments included a quite diverse set of tasks (including validation set perplexities, downstream task accuracies, instruction tuning, etc).

**Additional Feedback:**

Typos:

1. I think the "bottom right" and "bottom left" are flipped when the y-axis is being explained in Figure 1 caption.

2. Line 119 has 2 "froms" in "come from from"

**Clarity:**

A few parts of the paper's writing are quite confusing.

1. In Figure 1, it is quite difficult to understand what the x-axis means. What are these multipliers referring to? Is there a concrete number that we can anchor the multipliers on (such as 1.00x = 10B)? Also, this figure is 2 full pages ahead of where it is mentioned in text.

2. In section 3.4, the authors pointed out a few common problems with the two existing methods, and claimed that these common problems are the motivation of the new method that sequentially applies the two existing methods. It is unclear how sequentially applying 2 methods that have the same shortcomings can overcome the shortcoming.

**Correctness:**

There are a few parts of the experiment procedures where the correctness/soundness are questionable.

1. For the experiment in fixed compute regime, it is unclear what exact Dsource value is used in this experiment since the charts in Figure 1 only used relative measures between different Dsource sizes. Also, since D4 reduces the overall data size by Rdedup*Rproto while SemDeDup only reduces by Rdedup and SSL prototype only reduces by Rproto, the experiment seems unfair since we are D4 effectively is allowed to have a smaller overall R than either SemDedup or SSL Prototype alone.

2. D4 only marginally outperforms the two existing methods (SemDeDup and SSL Prototype) in the first experiment, and then the two existing methods was not included in any of the other experiments. Thus, there is not enough evidence showing that D4 (which sequentially applying the two existing methods with a re-clustering in-between) is significantly better than applying either existing methods alone. The highlighted "2% performance gain and 20% more data efficiency" is compared against random data selection instead of any existing data selection methods.

3. It is unclear why the particular D4 setup is the best. Since the paper shows that reclustering is essential, why can't we just take one of the methods, apply it twice with a re-clustering in-between (i.e. SemDeDup + Recluster in embedding space + SemDeDup, or  SSL Prototype + re-cluster + SSL Prototype)? Can we do SSL Prototype first then SemDeDup? There is no experiments showing the superiority of D4 over any of these setups, nor was there any explanations on why these other setups are not viable.

**Documentation:**

No new datasets.

**Ethics:**

No ethical concerns.

**Limitations:**

The limitations are adequately discussed at the end of the paper. There are no clear negative societal impacts of this work.

**Opportunities For Improvement:**

1. This paper lacks novelty or significance in contribution. The new proposed method, D4, sequentially composes two existing data deduplication methods, and there is no experiment showing significant improvement in performance of D4 over either methods.

2. The paper lacked some important comparisons in its experiments. Since D4 sequentially composes two existing methods, it is crucial to show that combining the methods created a very significant gain in performance over either methods alone. However, in the paper, D4 only showed marginal gains over either methods in the first experiment, and the two existing methods were not compared to in any of the later experiments. This puts the significance of D4 into question.

3. The presentation of the paper can be improved. It is unclear what the x-axis of Figure 1 means. The description of the motivation behind D4 is confusing. See clarity section for details.

**Relation To Prior Work:**

The novelty of this work is not clear, as the new proposed method is just applying 2 existing data deduplication sequentially (SemDedup + K-means-reclustering + SSL prototype) and there is no convincing evidence that the proposed method is significantly better than either existing method.

**Summary And Contributions:**

The authors proposed a new data selection strategy (D4) for deduplicating training data for large language models. The method involves sequentially applying two existing methods, SemDedup and SSL Prototypes, with a re-clustering step in-between. The authors conducted experiments to demonstrate that D4 outperforms random data selection by 2% in performance and 20% in training efficiency, as well as marginally outperforming SemDedup and SSL Prototypes applied alone. The authors also performed experiments to show the necessity of the re-clustering steps.

---

> ### Author Response · Authors · 2023-08-21
> **Response Part 1**
>
> Dear Reviewer qR3n,
>
> Thank you for your detailed review, and for the many insightful comments and suggestions. We are glad that you agree that improving data efficiency of pre-training is an important question, and find our experimental / evaluation setup diverse. Please find responses to your suggestions below (we have broken the reply up into multiple responses due to the character limit)
>
> > This paper lacks novelty or significance in contribution.
>
>
> We understand and appreciate the reviewer’s concern regarding the novelty of our work. We want to provide some additional context that we believe is important.
>
> (1) The current state-of-the-art method for reducing the redundancy of web-crawled data for LLM pre-training is MinHash-based fuzzy deduping, used in many recent data pipelines such as LLaMa (Touvron et al. 2023), RefinedWeb (Penedo et al. 2023), MassiveWeb (Rae et al., 2022), and The Pile (Gao et al., 2020). SSL Prototypes have never been used in the language domain, and the impact of SemDeDup is only demonstrated using relatively small models (125M) and on a dataset that has very naive deduplication in the first place (C4, no MinHash de-dup). Our work is the first to successfully demonstrate the benefits of data selection at the 6.7B scale for LLM pre-training.
>
> (2) We believe that the LLM community currently could benefit from more scientific rigor when it comes to data curation. Details about data are often not published, but even when they are, reports lack experiments that would ablate the effect of particular choices regarding the training data (we refer the reviewer to the introduction of [2] for more context on this). This work is one of the first to systematically study the effect of data on large-scale pre-training. We use a significant compute budget (on the order of magnitude of 0.5 million GPU hours, which is ~1-1.5 million dollars at current market GPU rates*), and publish all details for different data setups we use. We believe that by doing open, and scientifically rigorous explorations on the effect of data on pre-training, we have delivered actionable and useful insights for LLM practitioners that would otherwise go undocumented.
>
> (3) The adaptation of the individual methods to the domain of LLM pre-training is not a trivial process for various reasons, starting from the engineering challenges of scaling selection metric inference to hundreds of billions of tokens of web-text, to the intricacies of evaluating LLM performance under training distribution shift (Section 4.4.1). One of the key reasons adapting these methods to LLM pre-training is difficult due to scale (e.g. we have to embed, cluster, and find nearest neighbors for terabytes of text data).
>
> (4) To contextualize our improvements in downstream accuracy, we would like to refer to Figure 3 of the OPT paper [1]. This figure shows that a 2% improvement in downstream task accuracy is about as much as (1) the maximum difference one can observe between the performance of OPT and GPT3 at any given scale and (2) how much both GPT3 and OPT improve between 2.7B and 6.7B model scales.
>
> In summary, we believe that the introduction of D4 is not our work's sole (or perhaps not even the main) novelty. Novelty also lies in applying existing methods in a new domain and in our rigorous analysis. Furthermore, our results are very practically relevant: for example, the 20% efficiency gains for 6.7B model scale directly translates to 20% cost savings (which translates to up to $21,000 saved*). These contributions together resulted in the presentation of a new data selection method that significantly outperforms the currently prevailing approach in the community.
>
> Footnotes:
>
> *Current market rates for A100 80GB GPUs vary between ~$ 2-5 / hour (per gpu)
>
> Citations:
>
> [1]: Zhang, Susan, et al. "Opt: Open pre-trained transformer language models." arXiv preprint arXiv:2205.01068 (2022).
>
> [2]: A Pretrainer’s Guide to Training Data: Measuring the Effects of Data Age, Domain Coverage, Quality, & Toxicity

---

> > ### Author Response · Authors · 2023-08-21
> > **Response Part 2**
> >
> > > It is crucial to show that combining the methods created a very significant gain in performance over either methods alone.
> >
> > We completely agree with the reviewer that it is important to show that combining the methods creates significant gains. To provide some additional evidence, we ran an experiment using a different training dataset (C4) and for the fixed-data regime (e.g. repeating data) at the 125M scale. These are shown in Figure A20 and Figure A16 respectively.  We want to highlight that at highest selection ratios, D4 noticeably outperforms other methods. This is significant because high selection ratios correspond to aggressive source dataset pruning e.g. when selection methods are forced to prioritize which data points they want to keep (at low selection ratios, most selection methods are likely to have substantial overlap (see Figure A5) since we are barely pruning the source dataset).
> >
> > While we agree with the reviewer that 6.7B scale experiments demonstrating the benefits of D4 over individual methods would have provided stronger evidence, we also had to balance our compute budget very carefully such that we could provide a thorough scientific analysis of the methods we use but also produce results at a model scale that makes our work relevant to LLM practitioners who concern themselves with LLMs above the 7B parameter scale. This motivates our particular experimental setup for Figures 1 and 2. We also refer the reviewer to point (4) from our comments above regarding downstream task accuracy in order to contextualize the downstream task accuracy plot of Figure 1.
> >
> >
> > >  It is unclear what the x-axis of Figure 1 means.
> >
> > Thanks for pointing this out, we agree that the x-axis for this figure is confusing. We have changed the x-axis to one concrete number — the selection ratio, which is exactly the % of documents chosen from the source dataset. We have updated the definition in the “Notation” section of Section 3.
> >  There is no exact number for the size of the source dataset for a given multiplier (or selection ratio), because different documents have different # of tokens, and the training budget is set in terms of # tokens. For example, if I choose only long documents, I will need a much smaller source dataset to saturate a particular training budget. For that reason, we hesitate to say that particular ticks on the x-axis correspond to particular source dataset sizes; instead, we now note in the caption (now bolded in the caption) that when we decrease R, we increase the source dataset size  (e.g. we choose 1/4 of documents from a 4x’ed size datasets) so that we can still select enough documents to saturate the training token budget.
> > Note: we have swapped Figure 1 and Figure 2 so that the text is closer to the actual figures, so these changes are reflected in Figure 2.
> >
> > > The experiment seems unfair since we are D4 effectively is allowed to have a smaller overall R
> >
> > The reviewer notes correctly that the sequential composition of two filtering steps in D4 results in an overall smaller selection ratio R. We do account for this in all the x-axes for our figures, however. To compute the values used for the x-axis, we use R=R_proto or R=R_dedup for individual methods and R = R_proto * R_dedup for D4, so if we are trying to select data with D4 for R = 0.5, we could do R_dedup = 0.75 and R_proto = 0.6666666. Throughout the paper, we use R_dedup = 0.75, and vary R_proto to obtain specific values of R.
> >
> > > It is unclear why the particular D4 setup is the best.
> >
> > We chose to test the SemDeDup -> SSL Proto sequencing to solve a particular problem that we identified and we have also run ablations to measure the importance of the reclustering step in our algorithm. Still, the reviewer makes a great suggestion that we could have used alternative combinations of the two individual methods as baselines in order to put our intuition to test. While the SemDeDup -> SemDeDup combination should be mathematically equivalent to a single run of SemDeDup (the clustering step is not involved in the actual selection process, it is only an implementation detail that speeds up computation), a combination such as SSL Proto -> SSL Proto could possibly be a reasonable alternative. We unfortunately do not have available compute to finish all combinations in time for the rebuttal period, but we will try our best to add it to the final version of the paper.
> >
> >
> > > In Figure 1, it is quite difficult to understand what the x-axis means
> >
> > See comment above about Figure 1 x-axis. Note: we have swapped Figure 1 and Figure 2 so that the text is closer to the actual figures, so these changes are reflected in Figure 2.

---

> > > ### Author Response · Authors · 2023-08-21
> > > **Response Part 3**
> > >
> > > > It is unclear how sequentially applying 2 methods that have the same shortcomings can overcome the shortcoming
> > >
> > > We apologize for the confusion that our wording regarding the motivation for D4 has caused - the two methods do not have the same shortcomings. What is a shortcoming in SSL Prototypes (being affected by clusters that overfit on highly redundant repeated data) happens to be exactly what SemDeDup is designed to solve (that is, to remove that duplication within those clusters). This finding leads us to the introduction of D4, that is SSL Prototypes preceded by the application SemDeDup and a reclustering step. We have adjusted the wording in Sections 1 and 3.4 of the revised manuscript in order to better convey this message.
> > >
> > > We thank the reviewer again for their time reviewing our work. We hope that the revised figures and explanations we added following the reviewer’s suggestions helped to clear up some of the questions regarding our work. If this is the case, we respectfully ask the reviewer to increase their rating accordingly.

---

> > > > ### Comment · Reviewer_qR3n · 2023-08-23
> > > >
> > > > Thank you for your comprehensive and detailed response! I believe the authors have sufficiently addressed my concerns, especially about novelty and significance. I raised my score to 6.

---

### Official Review · Reviewer_QH9f · 2023-07-28
**Accept**

**Rating:** 8
**Confidence:** 5
**Correctness:** Yes
**Clarity:** Yes

**Strengths:**

1. This is a very clearly written paper, motivations are explained well.
2. Related work discussion is thorough, including the idea of data selection from non-text domains.


**Additional Feedback:**

-

**Documentation:**

Yes

**Limitations:**

.

**Opportunities For Improvement:**

.

**Relation To Prior Work:**

Yes

**Summary And Contributions:**

This paper explores the effect of clever data selection on pre-training and downstream performance of LLMs. The results are insightful:
1. They show that careful data selection (on top of de-duplicated data) via pre-trained model embeddings can speed up training significantly and improves average downstream accuracy on multiple tasks at the 6.7B model scale.
2. They also show that repeating data intelligently consistently outperforms baseline training, and repeating random data has the opposite effect. Repeating data via epoching can be beneficial for LLM training.
3. In general, the authors question the paradigm of randomly sampling “more” web data (not necessarily the “right” data)
4. The paper proposes a new data strategy “D4” to overcome the impact of duplicate-driven clusters (in SSL Prototypes and SemDeDup methods) in the embedding space on performance.

---

> ### Author Response · Authors · 2023-08-21
>
> Dear reviewer QH9f,
>
> Thank you for taking the time to review our work. We are glad to hear that you found that our motivations and work explained well! You summarized the key take away from our work excellently: we question the current standard in LLM pre-training of sampling “more” but not necessarily the ”right” data. We hope that our work motivates others to explore the effect of data curation in large-scale LLM pre-training.

---

### Official Review · Reviewer_Wz3a · 2023-07-28
**Review for paper "D4: Improving LLM Pretraining via Document De-Duplication and Diversification"**

**Rating:** 6
**Confidence:** 2
**Correctness:** Yes
**Clarity:** Yes

**Strengths:**

1. This paper evaluated different data selection strategies for standard LLM pre-training setups where data has already been manually filtered or de-duplicated.
2. The topic raised in this paper is interesting and the paper is well written.
3. The proposed approach may have more applications and impact.

**Additional Feedback:**

NA

**Documentation:**

NA

**Limitations:**

1. In evaluation part, the authors evaluated model with 6.7B parameters trained on 100B tokens. It will be better to add more experiment for larger models, for example models with 100B parameters.
2. We know that the quality of the embedding space and clustering is crucial to the performance of data selection methods. It will be better to explore the effect of embedding space on data selection.
3. The data distribution of selected samples and training samples may different. It will be better to add more demonstration for the efficacy of D4 on a mix of training distributions.
4. For training sample selection, it will be better to add one more reference paper: "At the speed of sound: Efficient audio scene classification" which discuss how to assign weights for different training samples.

**Opportunities For Improvement:**

1. In evaluation part, the authors evaluated model with 6.7B parameters trained on 100B tokens. It will be better to add more experiment for larger models, for example models with 100B parameters.
2. We know that the quality of the embedding space and clustering is crucial to the performance of data selection methods. It will be better to explore the effect of embedding space on data selection.
3. The data distribution of selected samples and training samples may different. It will be better to add more demonstration for the efficacy of D4 on a mix of training distributions.
4. For training sample selection, it will be better to add one more reference paper: "At the speed of sound: Efficient audio scene classification" which discuss how to assign weights for different training samples.
typos:
line 291: comon → common


**Relation To Prior Work:**

Yes

**Summary And Contributions:**

This paper proposed D4, a method for data curation on LLMs that improves training efficiency by 20% across multiple model scales, with larger gains at increased model scale. The authors also demonstrated that, in contrast to common practice, repeating data via epoch can be beneficial for LLM training, but only if the data subset is intelligently selected.
1. This paper evaluated different data selection strategies for standard LLM pre-training setups where data has already been manually filtered or de-duplicated.
2. The topic raised in this paper is interesting and the paper is well written.
3. The proposed approach may have more applications and impact.

---

> ### Author Response · Authors · 2023-08-21
> **Response to Reviewer**
>
> Dear reviewer Wz3a,
>
> Thank you for your time reviewing our work. We are glad that you found our work interesting, impactful, and well-written! Please find our responses to your comments below:
>
> > It will be better to add more experiment for larger models
>
> We fully agree with the reviewer that it would be exciting and useful to see the benefits D4 brings for 100B+ LLM training runs. Figure 4 and Figure A2 suggest that bigger models / longer training runs could benefit even more and might exhibit even larger gains. As such, we would expect significant efficiency gains at the 100B+ scale. Unfortunately our compute budget did not allow for larger experiments: pre-training a single 100B OPT model on 2T tokens* would cost roughly 5 million dollars, even at the current lowest market rate for A100 80GB GPUs ($2 / hour). Our 6.7B OPT pre-training runs alone cost roughly 0.5 million dollars. We do hope to see our method applied for models at the 100B+ parameter scale, and leave it as future work.
>
> > It will be better to explore the effect of embedding space on data selection.
>
> We have taken the reviewer’s comment into consideration and have run additional experiments to explore the role of the embedding space in the efficiency of our methods. We have added our new results in Appendix A.8 of the revised manuscript. In summary, we find that the OPT model embedding space we use works best for large models (6.7B) and that Sentence BERT embedding spaces work best for smaller models (125M / 1.3B).
>
> >  It will be better to add more demonstrations for the efficacy of D4 on a mix of training distributions.
>
> We agree with the reviewer that applying D4 on a mix of training distributions is indeed interesting, but we are limited by compute budget since pre-training models on large token budgets is expensive (our current 6.7B pre-training runs on 100B tokens alone cost ~0.5 million dollars). We leave this as future work.
>
> > For training sample selection, it will be better to add one more reference paper
>
> We thank the reviewer for this suggestion, we have included this reference in Section 2.
>
>
> We hope that the reviewer found our responses, additional experiments, and revisions to the manuscript helpful. If this was the case, we ask the reviewer to kindly consider increasing the review score.
>
> ### Footnotes:
> * We chose 2T for this calculation to continue following compute-optimal scaling laws provided by [1], as we did in the rest of the work. More specifically, we look at Table 3 in [1] and estimate 2T tokens as a lower bound for how many tokens we would need to train a 100B OPT model to remain in line with the compute-optimal scaling laws.
>
> ### Citations:
> [1]: Hoffmann, Jordan, et al. "Training compute-optimal large language models." arXiv preprint arXiv:2203.15556 (2022).

---

### Decision · Program_Chairs · 2023-09-22

**Decision:**

Accept (Poster)

**Comment:**

The paper has merits; all reviewers vote for borderline accept. The experiments and results are solid. This paper shows that deduplicating a pretraining corpus improves pretraining. There are a few small concerns, such as: the authors use cc_net output directly, whereas ccn_net output itself is very noisy, and some of the issues with high number of semantic duplicates might go away if they dedup after cleaning. Overall, I vote for weak accept.